# CTCF controls three-dimensional enhancer network underlying the inflammatory response of bone marrow-derived dendritic cells

Bobae Yang[1,2], Sueun Kim[1,2], Woong-Jae Jung[1], Kyungwoo Kim[1,2], Sugyung Kim[1,2], Yong-Jin Kim[1,2], Tae-Gyun Kim[1,2], Eun-Chong Lee[1], Jung-Sik Joo[1,2], Chae Gyu Park[2,3], Sumin Oh[4], Kyung Hyun Yoo[4] & Hyoung-Pyo Kim[1,2,5] ✉

Dendritic cells are antigen-presenting cells orchestrating innate and adaptive immunity. The crucial role of transcription factors and histone modifications in the transcriptional regulation of dendritic cells has been extensively studied. However, it is not been well understood whether and how three-dimensional chromatin folding controls gene expression in dendritic cells. Here we demonstrate that activation of bone marrow-derived dendritic cells induces extensive reprogramming of chromatin looping as well as enhancer activity, both of which are implicated in the dynamic changes in gene expression. Interestingly, depletion of CTCF attenuates GM-CSF-mediated JAK2/STAT5 signaling, resulting in defective NF-κB activation. Moreover, CTCF is necessary for establishing NF-κB-dependent chromatin interactions and maximal expression of pro-inflammatory cytokines, which prime Th1 and Th17 cell differentiation. Collectively, our study provides mechanistic insights into how three-dimensional enhancer networks control gene expression during bone marrow-derived dendritic cells activation, and offers an integrative view of the complex activities of CTCF in the inflammatory response of bone marrow-derived dendritic cells.

Gene expression in eukaryotic cells is controlled by DNA sequence-specific transcription factors, which act by binding to gene regulatory elements such as promoters, enhancers, and boundary elements. Enhancers often exist at a considerable distance from their target genes, and chromatin looping provides a mechanism for distal enhancers to interact with their target promoters[1–3]. Due to technological advancements, recent studies on the structural organization of the genome have revealed that the gene regulatory mechanism is tightly linked to the multi-layered chromatin conformation[4–6]. Mammalian genomes are folded into a complex three-dimensional (3D) chromatin architecture comprising compartments, topologically associating domains (TADs), and chromatin

[1]Department of Environmental Medical Biology, Institute of Tropical Medicine, Yonsei University College of Medicine, 50-1 Yonsei-ro, Seodaemun-gu, Seoul 03722, Korea. [2]Brain Korea 21 PLUS Project for Medical Science, Yonsei University College of Medicine, 50-1 Yonsei-ro, Seodaemun-gu, Seoul 03722, Korea. [3]Laboratory of Immunology, Severance Biomedical Science Institute, Yonsei University College of Medicine, 50-1 Yonsei-ro, Seodaemun-gu, Seoul 03722, Korea. [4]Laboratory of Biomedical Genomics, Department of Biological Sciences, Research Institute of Women's Health, Sookmyung Women's University, Seoul 04310, Korea. [5]Division of biology, Pohang University of Science and Technology, Pohang 37673, Korea. ✉e-mail: kimhp@yuhs.ac

loops[7]. Individual chromosomes segregate into active "A" or inactive "B" compartments, which may arise as a consequence of homotypic interactions between genomic regions with similar transcription and chromatin states[7]. TADs divide the mammalian genome at the sub-megabase level and display preferential intra-domain chromatin interactions[4,5]. CTCF is a well-known transcription factor with 11 zinc finger domains that defines TADs by binding to boundary elements and promotes the formation of chromatin loops along with cohesin proteins[8,9]. The spatial insulation imposed by CTCF and TAD boundaries has been proposed to influence gene expression by facilitating proper intra-domain enhancer-promoter communications while inhibiting cross-boundary enhancer action to prevent inappropriate gene activation[3,10–13].

Dendritic cells (DCs) are professional antigen-presenting cells that provide critical signals for eliciting innate and adaptive immune responses[14]. Upon detecting invading pathogens or other inflammatory stimuli, DCs respond with robust transcriptional and epigenetic reprogramming, directing a series of maturation events such as antigen uptake, antigen processing, and migration to specialized lymphoid organs[15–17]. Thereafter, activated DCs modulate lymphocyte activation and differentiation by upregulating the expression of major histocompatibility complexes, co-stimulatory molecules, and a variety of pro-inflammatory cytokines[18,19]. These responses are initiated through a set of innate pattern recognition receptors, such as toll-like receptors (TLRs), which discriminate harmful foreign materials from ourselves and are propagated through well-characterized signaling pathways[20,21]. However, an aberrant activation of DCs can also lead to diverse autoimmune and immune-mediated inflammatory diseases such as multiple sclerosis[22,23]. Therefore, understanding the molecular mechanisms underlying the activation of DCs is important for developing strategies against inflammatory disorders.

Previously, we reported that CTCF is required for the homeostasis of epidermal Langerhans cells and bone marrow (BM) primitive hematopoietic stem cells[24,25]. However, the mechanism through which CTCF regulates the complex molecular events during DC activation has not yet been comprehensively investigated. Here, we analyze the global transcriptional, epigenetic, and topological changes occurring in mouse bone marrow-derived DCs (BMDCs) in response to pathogenic stimuli and demonstrate that CTCF is essential in shaping the three-dimensional enhancer networks for the optimal immune function of BMDCs.

## Results

### Disrupted occupancies of CTCF and SMC1 due to CTCF depletion in in vitro-generated GM-CSF dendritic cells

To decipher the role of CTCF in the gene expression and immunomodulatory function of the primary mouse DCs, we used common in vitro DC differentiation models, where cultures of naïve bone marrow (BM) in the presence of GM-CSF leads to differentiation of precursors into a monocyte-derived DC state[26] (denoted from here on as BMDC). We added 4-hydroxytamoxifen (4-OHT) to the BM cultures on the first day of differentiation to remove loxP-flanked Ctcf alleles for Ctcf conditional knockout cells (CreER;CTCF^fl/fl) but not for WT cells (CreER;CTCF^wt/wt). Flow cytometry analysis demonstrated that GM-CSF BM cultures comprised mainly CD11c⁺MHCII^high CD11b^int CD115⁻CD135⁺ GM-DCs along with a minor fraction of CD11c⁺MHCII^int CD11b^high cells that might have potential to differentiate into GM-DCs (Supplementary Fig. 1)[27–29]. Following in vitro differentiation, BMDCs were activated using the bacterial cell wall component LPS to trigger an inducible gene expression program (Fig. 1a).

Depletion of endogenous CTCF in the BMDCs was confirmed at the mRNA and protein levels (Fig. 1b, c). Genome-wide CTCF binding patterns, measured by ChIP-seq, in untreated wild-type BMDCs (WT) were mostly similar to those in LPS-stimulated wild-type BMDCs (WTL), indicating that LPS treatment by itself does not affect overall CTCF binding (Fig. 1d, e, Supplementary Fig. 2a). In contrast, CTCF occupancy was lost or severely reduced at all of its binding sites in CTCF-deficient BMDCs, in both unstimulated and LPS-stimulated cells (KO and KOL, respectively) (Fig. 1d, e, Supplementary Fig. 2a). Genome-wide occupancy of SMC1, a subunit of the cohesin complex functionally associated with CTCF to mediate long-range chromatin interaction, demonstrated a similar pattern to that of CTCF; it was not largely affected by LPS stimulation but was lost or considerably reduced due to CTCF depletion (Fig. 1d, e, Supplementary Fig. 2b).

During GM-CSF-mediated differentiation, compared with WT BM, CTCF-deficient BM demonstrated a similar apoptosis profile but had slightly reduced proliferation (Supplementary Fig. 3a and b). The percentage of CD11b⁺CD11c⁺ DCs after 6 days of GM-CSF-supplemented culturing was comparable between WT and CTCF-deficient BMs (Supplementary Fig. 1 and 3c). However, the resultant number of CTCF-deficient BMDCs decreased by 50% compared with that of WT BMDCs (Supplementary Fig. 3d), possibly due to a slower growth rate in CTCF-deficient BMDCs. Moreover, CTCF-deficient BMDCs exhibited equivalent capacity for antigen uptake and induction of T cell proliferation, and relatively higher LPS-stimulated activation of costimulatory molecules compared with WT BMDCs (Supplementary Fig. 3e–h).

### TAD insulation was compromised by CTCF depletion but not by LPS stimulation

To investigate the extent to which LPS stimulation affects higher-order chromatin architecture of BMDCs in the presence or absence of CTCF, we performed in situ Hi-C, high-throughput 3C-based experiments quantifying the genome-wide DNA contacts[3]. Contact maps (Supplementary Fig. 4a) and compartment signals (Fig. 2a and Supplementary Fig. 4b) demonstrated that segregation of active and inactive chromosome domains into A and B compartments was not significantly affected by LPS stimulation (WT vs. WTL; $r^2 = 0.99$) or CTCF depletion (WT vs. KO; $r^2 = 0.93$ when untreated, and WTL vs. KOL; $r^2 = 0.94$ when LPS-stimulated). We next defined TADs using insulation scores and found that the number and size of TADs were not affected by LPS stimulation; however, fewer and larger TADs were observed in CTCF-deficient BMDCs compared with WT BMDCs (Fig. 2b, c). Moreover, the capacity of preventing inter-TAD interactions, represented by the TAD boundary strength, was significantly reduced by CTCF depletion, but not by LPS stimulation (Fig. 2d). The data suggest that CTCF is dispensable for higher order chromatin compartment but essential for TAD organization, which was consistent with previous reports[30–32], and that the activation of BMDCs by LPS stimulation has little effect on the three-dimensional genome organization at the resolution of compartment and TAD.

### CTCF is required for the maintenance of enhancer-centric chromatin interactions

Further, we explored intra-TAD chromatin looping, focusing on enhancer-promoter interactions across the genome, by performing H3K27ac HiChIP to generate high-resolution contact maps of active enhancers and target genes in BMDCs. Statistically significant H3K27ac-based chromatin interactions were called using FitHiChIP at a resolution of 10 kb, with FDR < 10⁻⁵, a minimum genomic distance of 20 kb and a maximum genomic distance of 2 Mb. For untreated (WT) and LPS-stimulated (WTL) BMDCs, we identified comparable number of high-confidence H3K27ac HiChIP loops (70612 in WT versus 62373 in WTL) (Fig. 2e), most of which enriched CTCF and SMC1 at least at one of the loop anchors (Supplementary Fig. 5a and b). In contrast, CTCF depletion dramatically decreased the number of H3K27ac HiChIP loops (10466 in KO and 5088 in KOL), suggesting a critical role of CTCF in the maintenance of enhancer-centric chromatin interactions at a sub-TAD scale (Fig. 2e). Further analysis of chromatin

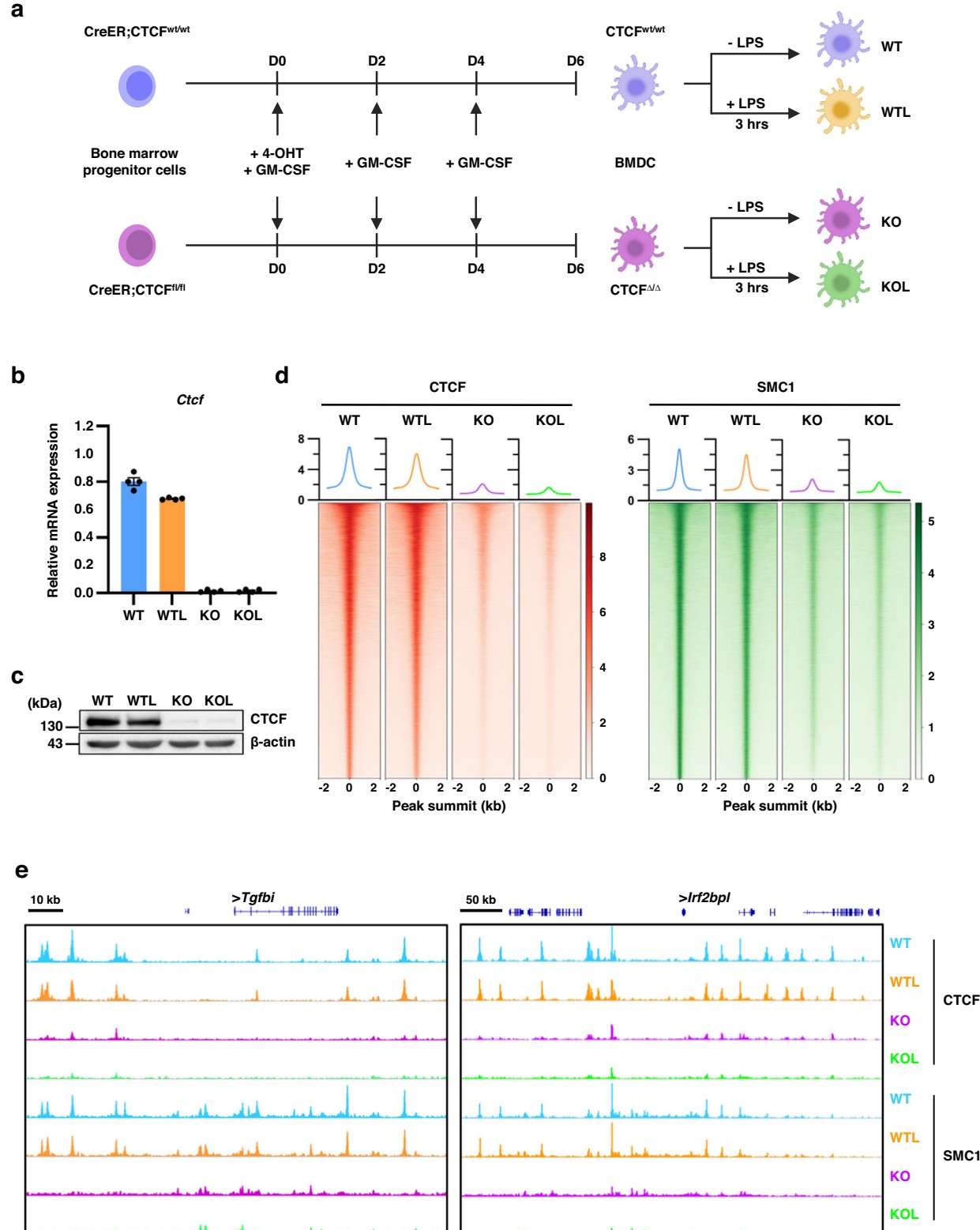

interactions demonstrated that the major fraction of the H3K27ac HiChIP loops (~50% in WT and WTL, ~75% in KO and KOL) represented chromatin interactions connecting two gene regulatory elements, such as promoter–promoter, promoter–enhancer, or enhancer-enhancer (Fig. 2f). In addition, each promoter can interact with a single enhancer or multiple enhancers acting in concert (Supplementary Fig. 5c), while multiple promoters can be regulated by a single enhancer (Supplementary Fig. 5d). Moreover, a significant fraction of the chromatin interactions (5% in WT and WTL; 15% in KO and KOL) was found between promoter pairs and many promoters can interact with multiple other promoters (Fig. 2f and Supplementary Fig. 5e), suggesting a potential regulatory function in distal gene regulation.

**Fig. 1 | Disrupted occupancies of CTCF and SMC1 due to CTCF depletion in BMDCs. a** Schematic showing in vitro DC differentiation models, where cultures of naïve BM in the presence of GM-CSF leads to differentiation of precursors into BMDCs. We added 4-hydroxytamoxifen (4-OHT) to the BM cultures on the first day of differentiation to remove loxP-flanked Ctcf alleles for Ctcf conditional knockout cells (CreER;CTCF[fl/fl]) but not for WT cells (CreER;CTCF[wt/wt]). Following in vitro differentiation, WT and CTCF-deficient BMDCs were activated using LPS. WT: untreated wild-type BMDC, WTL: wild-type BMDC treated with LPS for 3 h, KO: untreated CTCF knock-out BMDC, KOL: CTCF knock-out BMDC treated with LPS for 3 h. **b**, **c** Efficient depletion of CTCF at the mRNA (**b**) and protein (**c**) levels in CTCF-deficient BMDCs. Error bars represent mean ± standard error of the mean (s.e.m.). $n = 2$ biologically independent samples were used for each group. **d** Heatmaps of ChIP-Seq signal called for CTCF (left) and SMC1 (right) showing disrupted global occupancy of each protein in CTCF-deficient BMDCs. Histogram showing the average tag density of CTCF or SMC1 ChIP-seq peaks are displayed on top of each heatmap. **e** Snapshot of ChIP-seq signal tracks for CTCF and SMC1 in the representative genomic region. Source data are provided as a Source Data file.

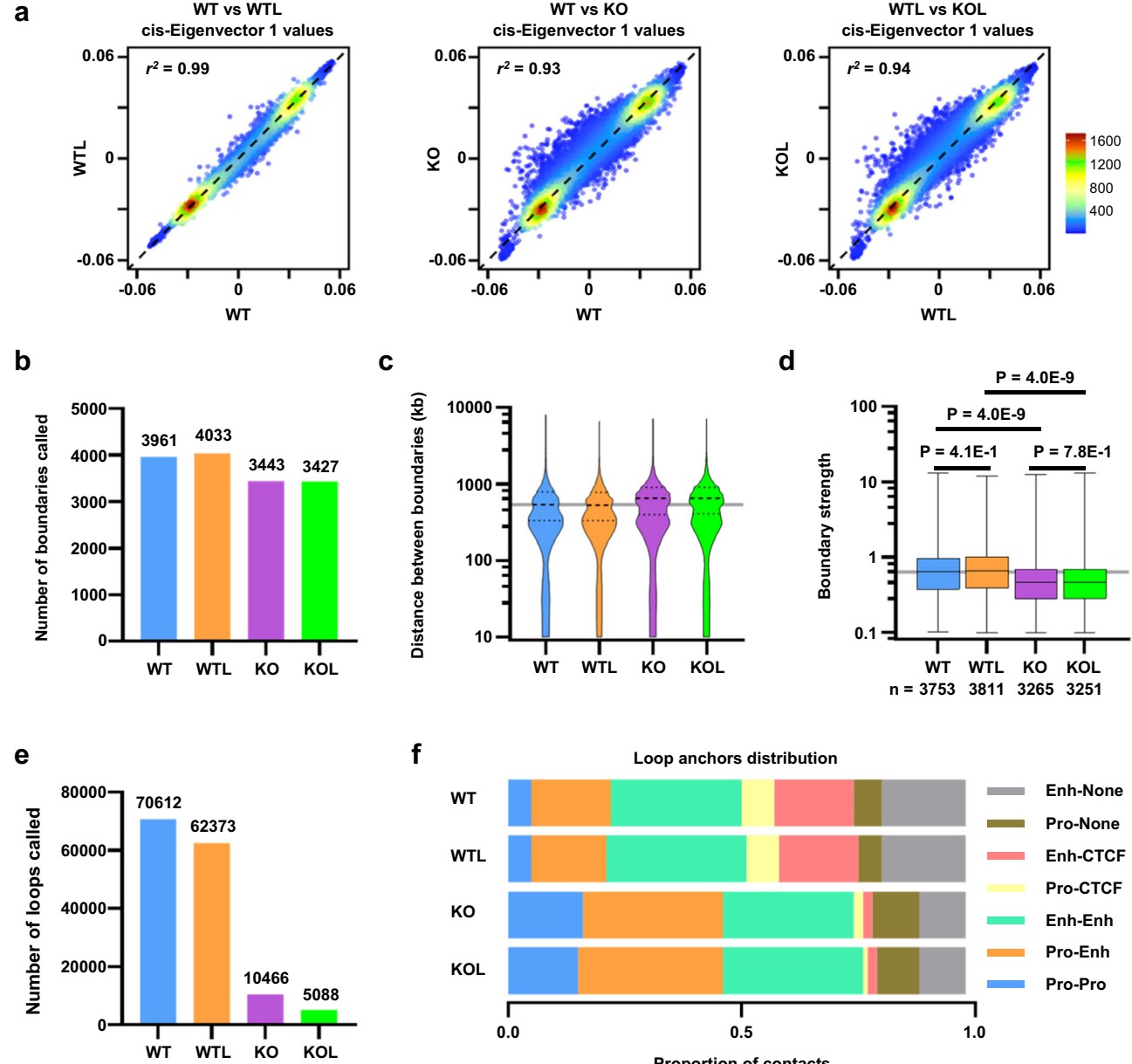

**Fig. 2 | Insulation of TAD and enhancer-promoter looping were reduced by CTCF depletion in BMDCs. a** Either LPS stimulation or CTCF depletion does not significantly affect higher-order segregation of A and B compartments when evaluated by cis-Eigenvector 1 values. **b–d** CTCF is required for proper insulation of TAD in BMDCs. Number of boundaries (**b**), distance between boundaries (**c**), and boundary strength (**d**) obtained with Hi-C data. Box-plot with midline = median, box limits = Q1 (25th percentile)/Q3 (75th percentile), whiskers = minimum and maximum values. Significance was calculated using a one-way ANOVA with Tukey's multiple comparisons. The sample sizes ($n$) were labeled in the figure. **e**, **f** Depletion of CTCF disrupts enhancer-promoter looping in BMDCs. Number of loops (**e**) and distribution of regulatory elements at the anchors of the high-confidence H3K27ac HiChIP loops (**f**). Source data are provided as a Source Data file.

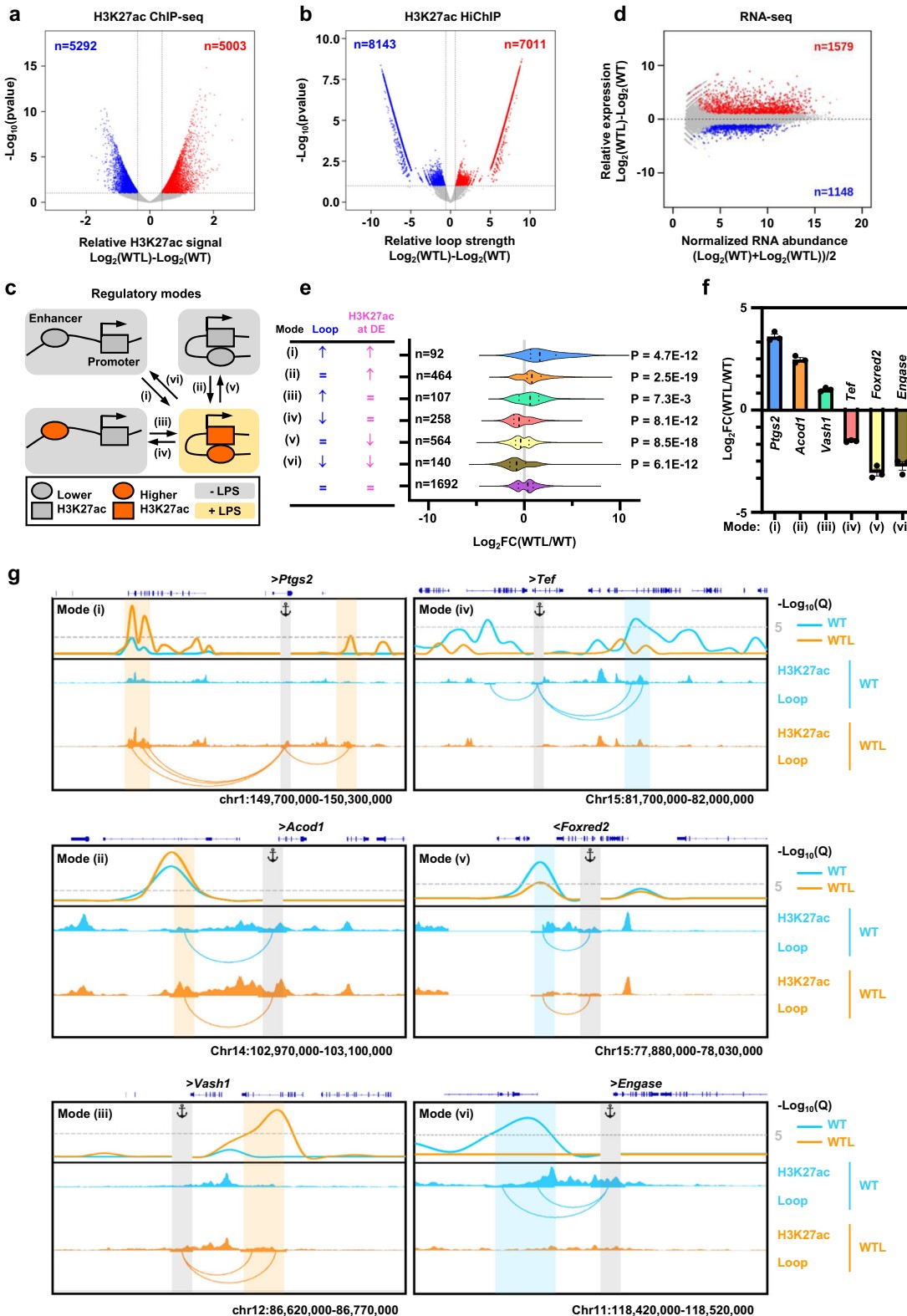

## Dynamic changes in loop strength and enhancer activity correlate with gene expression

A previous report demonstrated that LPS stimulation induced changes in the chromatin state of promoters and enhancers to control gene expression in mouse DCs[17]. Indeed, H3K27ac ChIP-seq analysis of untreated and LPS-stimulated BMDCs allowed us to characterize activation-inducible, activation-repressed, and constitutive enhancers on the basis of a differential H3K27ac level, which

serves as a proxy for enhancer activity, at the 10 kb-long loop anchors (Fig. 3a). Many enhancers are located quite far from the genes they regulate, and it is proposed that close physical proximity between distal enhancers (DE) and their target genes plays a critical role in the control of proper gene expression[33]. Thus, we investigated whether LPS stimulation mediates any dynamic changes in spatial proximity between enhancers and their cognate target promoters. Using differential loop analyses for H3K27ac HiChIP, we defined gain of loop,

**Fig. 3 | Dynamic changes in loop strength and distal enhancer activity control LPS-stimulated gene expression. a** Volcano plot showing significant LPS-mediated changes in H3K27ac level at the 10 kb-long loop anchors. The number of anchors exhibiting >1.3-fold increases in WT (blue) or WTL (red) BMDCs with a *p*-value < 0.1 has been indicated. **b** Volcano plot showing significant LPS-mediated changes in H3K27ac HiChIP loop strength. The number of loops exhibiting >1.5-fold increases in WT (blue) or WTL (red) BMDCs with a *p*-value < 0.1 has been indicated. **c** Schematic depiction of six possible regulatory modes (i-vi) controlling LPS-stimulated gene expression. **d** RNA-seq MA plot of WT versus WTL. The number of genes exhibiting >2-fold increases in WT (blue) or WTL (red) BMDCs with a false discovery rate <0.05 has been indicated. Significance in **a**, **b**, **d** was calculated using two-side Wald test by the nbinomWaldTest function in DESeq2. **e** Log2-fold changes in RNA expression for the genes corresponding to each regulatory mode shown in **c**. The genes associated with constant loops and constant distal enhancers were used as control. The number of genes in each regulatory mode with a false

discovery rate <0.05 has been indicated. The respective minimum and maximum values as well as the 25th, 50th, and 75th quartiles are shown for each violin plot. Significance was calculated using an unpaired two-tailed *t*-test. The sample sizes (*n*) were labeled in the figure. **f** Log2-fold changes in RNA expression for the typical genes representing each regulatory mode shown in **c**. Error bars represent mean ± standard error of the mean (s.e.m.). *n* = 3 biologically independent samples were used for each group. **g** Snapshot displaying dynamic changes in loop strength and enhancer activities for the typical genes representing each regulatory mode shown in **c**, **e** and **f**. Top: Virtual 4C plots (V4C) showing normalized H3K27ac HiChIP loop strength (represented as -Log10(*Q*)) with the TSS of each selected gene as viewpoint. Bottom: IGV browser showing ChIP-seq signal tracks for H3K27ac and arcs showing significant interactions of H3K27ac loops with −Log10(*Q*) ≥ 5. Only loops interacting with the viewpoint TSS were displayed. Source data are provided as a Source Data file.

loss of loop, and preformed constant loop following LPS stimulation (Fig. 3b). Thereafter, we considered six possible regulatory modes controlling LPS-stimulated gene expression, categorized by different combinations of dynamic changes in loop strength and distal enhancer (DE) activity: (i) gain of loop to activation-inducible enhancer; (ii) preformed loop with activation-inducible enhancer; (iii) gain of loop to constitutive enhancer; (iv) loss of loop from constitutive enhancer; (v) preformed loop with activation-repressed enhancer; and (vi) loss of loop from activation-repressed enhancer (Fig. 3c). Further, we performed RNA-seq to map changes in transcript abundance (Fig. 3d) and examined whether each regulatory mode shows any correlation with LPS-induced gene activation or repression (Fig. 3e). Compared to the control genes connected to constitutive distal enhancers by preformed constant loops, the genes connected to activation-inducible enhancers by gained loops (mode (i)) exhibited significantly upregulated gene expression, while genes connected to activation-repressed enhancers by lost loops (mode (vi)) demonstrated significantly downregulated gene expression (Fig. 3e). In addition, activation-inducible (mode (ii)) or activation-repressed enhancers (mode (v)) could significantly increase or decrease RNA expression, respectively, when these distal enhancers were connected to their target promoters by preformed constant loops (Fig. 3e). Moreover, acquisition (mode (iii)) or disruption (mode (iv)) of promoter interaction could activate or repress their target genes, respectively, even though LPS stimulation did not significantly change the enhancer activities at distal ends of the loops (Fig. 3e). Examples of dynamic changes in loop strength and enhancer activities corresponding to each regulatory mode and the resultant differential RNA expression of the target genes are shown in Fig. 3f, g. Overall, the results suggest that both spatial enhancer-promoter proximity and chromatin state of distal enhancer provide crucial regulatory mechanisms for the fine tuning of gene expression in response to LPS stimulation.

**Chromatin insulation imposed by CTCF constrains enhancer action to prevent aberrant gene expression**

Further, we explored the effect of CTCF depletion on the gene expression profile during the GM-CSF-mediated DC differentiation. Differential expression analysis of RNA-seq between WT and KO BMDCs revealed 4435 deregulated genes (FDR < 0.05, FC > 1.3), of which 2248 and 2187 were upregulated and downregulated, respectively, by CTCF depletion (Fig. 4a). Gene ontology analysis of upregulated genes in KO BMDCs revealed significant enrichment of genes known to be associated with metabolic processes, transport, and cell adhesion, while downregulated genes in KO BMDCs showed enrichment of genes associated with cell cycle, cell division, and immune system processes (Supplementary Fig. 6a and b).

Given that CTCF depletion weakened insulation capacity at the WT TAD boundaries but did not affect insulation capacity at

the KO TAD boundaries (Fig. 4b, c), we examined whether any enhancer-promoter interaction, prevented by strong TAD boundaries in WT BMDCs, could be established de novo within the same TADs in the CTCF-deficient BMDCs (Fig. 4d and Supplementary Fig. 6c). Indeed, 34% of the loops gained in CTCF-deficient BMDCs were observed across the WT boundaries (Fig. 4e); the expression of their target genes more likely upregulated, possibly due to the augmented spatial proximity between enhancer and promoter (123 upregulated vs. 53 downregulated; Fig. 4f and Supplementary Table 5).

The *Aldh1a2* gene, encoding aldehyde dehydrogenase 1A2 enzyme that catalyzes synthesis of retinoic acid from retinaldehyde[34], was one of the best examples for the upregulated gene expression caused by impaired TAD boundary integrity (Fig. 4f, g). The modest level of *Aldh1a2* gene expression in WT BMDCs was consistent with the observation that the promoter of *Aldh1a2* gene did not interact with a nearby super-enhancer (SE; 20–60 kb upstream) due to the strong insulation exerted by the TAD boundary between them. However, it interacted with distant CTCF binding regions (~300 kb downstream) which demonstrated very low levels of active H3K4me1 and H3K27ac markers but high enrichment of the repressive marker H3K27me3. Chromatin interaction between *Aldh1a2* promoter and distant CTCF-binding regions was not maintained in CTCF-deficient BMDCs (Fig. 4g). Instead, CTCF depletion allowed the *Aldh1a2* promoter to interact with the nearby SE as well as far upstream enhancers (denoted as E1 and E2) due to the disruption of a TAD boundary (Fig. 4g). Although H3K27ac levels of these newly-connected enhancers (E1, E2, and SE) did not changed significantly due to CTCF depletion, acquisition of close physical proximity between the *Aldh1a2* promoter and these constitutively active enhancers across WT TAD boundary seemed to be sufficient to drive increased *Aldh1a2* gene expression in CTCF-deficient BMDCs (Fig. 4g). The upregulated expression of *Aldh1a2* gene was further validated by higher H3K4me3 levels at the promoter (Fig. 4g) and increased mRNA and protein levels and enzyme activity in KO BMDCs (Fig. 4h–j). Given the critical role of retinoic acid in the induction and suppression of Treg and Th17 differentiation[35–37], respectively, enhanced expression of *Aldh1a2* in CTCF-deficient BMDCs might provide favorable conditions for the development of T cell tolerance.

CTCF could also insulate intra-TAD chromatin interactions and prevent inappropriate gene activation, given that 66% of gained loops after CTCF depletion were observed within WT TAD boundaries (Fig. 4e) and the expression of their target genes were more likely upregulated (300 upregulated vs. 125 downregulated; Supplementary Fig. 6d). For example, CTCF depletion resulted in the upregulation of *Chst3* gene expression due to the loss of CTCF-mediated insulation and thereby erroneously-established intra-TAD loops between constitutive distal enhancers and the *Chst3* promoter (Supplementary Fig. 6e and f). These results clearly demonstrated that CTCF plays a critical role in

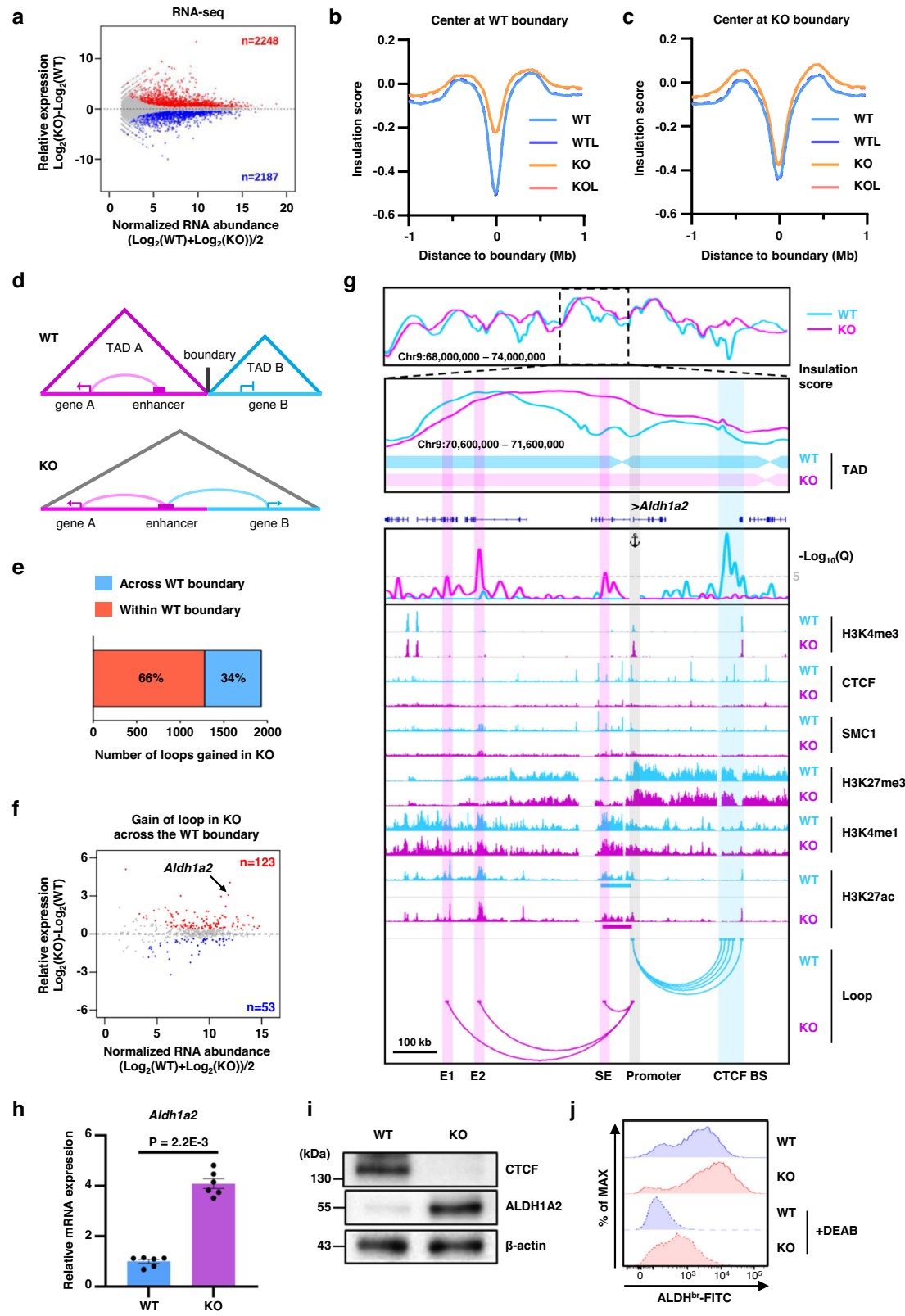

chromatin insulation and constrains enhancer action to prevent aberrant gene expression in mouse BMDCs.

**Attenuated JAK2/STAT5 signaling in CTCF-deficient BMDCs**
Given the importance of GM-CSF in the differentiation, proliferation, and survival of BMDCs[38], we evaluated whether CTCF depletion has any effect on the JAK/STAT pathway, one of the major signaling pathways triggered by the GM-CSF receptor. Western blot analysis demonstrated that GM-CSF induced phosphorylation of JAK2 and STAT5 in WT BMDCs, which was much weaker in KO BMDCs (Fig. 5a, b). In addition, ChIP-seq analysis with anti-STAT5 antibodies demonstrated that global enrichment of STAT5 was mostly lower in CTCF-deficient BMDCs than in WT BMDCs (Fig. 5c). Subsequently, we identified specific STAT5 peaks lost in KO BMDCs (n = 6714; Fig. 5d) and

**Fig. 4 | Disruption of TAD boundary in CTCF-deficient BMDCs leads to rewiring of enhancer-promoter interactions and dysregulation of RNA expression.** **a** RNA-seq MA plot of WT versus KO. The number of genes exhibiting >1.3-fold increases in WT (blue) or KO (red) BMDCs with a false discovery rate of less than 0.05 has been indicated. **b**, **c** Genome-wide averaged insulation plotted versus distance around insulation center at WT (**b**) or KO (**c**) TAD boundary. **d** A schematic showing that CTCF depletion leads to upregulation of inactive gene in WT via enhancer-promoter rewiring due to the TAD boundary disruption. **e** The number of loops gained in CTCF-deficient BMDCs which mediate enhancer-promoter interaction either across WT boundary or within WT boundary. **f** RNA-seq MA plot for the genes associated with "gain of loop across the WT boundary". The number of genes exhibiting >1.3-fold increases in WT (blue) or KO (red) BMDCs with a false discovery rate of less than 0.05 has been indicated. **g** Rewiring to already-active enhancers due to the disrupted TAD boundary in CTCF KO BMDCs leads to augmented *Aldh1a2* gene expression. Snapshot shows insulation score curves with TAD boundaries, virtual 4C plots, ChIP-seq signal tracks, and significant H3K27ac HiChIP loops (from top to bottom) at the *Aldh1a2* locus. Virtual 4C plots (V4C) shows normalized H3K27ac HiChIP loop strength (represented as -Log10($Q$)) with the TSS of *Aldh1a2* gene as the viewpoint. IGV browser shows ChIP-seq signal tracks for H3K4me3, CTCF, SMC1, H3K27me3, H3K4me1, and H3K27ac. Arcs shows significant interactions with −Log10($Q$) ≥ 5. Only loops interacting with the viewpoint were displayed. **h**, **i** Elevated expression of ALDH1A2 at the mRNA (**h**) and protein (**i**) level by CTCF depletion. Error bars represent mean ± standard error of the mean (s.e.m.). Significance was calculated using an unpaired two-tailed $t$-test using $n = 3$ independent samples. **j** Flow cytometry analysis of ALDH activity in WT and KO BMDCs using the Aldefluor™ assay in the absence or presence of the ALDH inhibitor DEAB. Source data are provided as a Source Data file.

overlapped them with promoters and/or distal enhancers to determine the candidate genes whose expression could be influenced by defective JAK2/STAT5 signaling ($n = 2403$; Fig. 5e). Differential RNA expression analysis of the genes associated with lost STAT5 binding revealed 971 genes deregulated due to CTCF depletion (FDR < 0.05, FC > 1.3), of which 495 and 476 were upregulated and downregulated, respectively (Fig. 5f). Interestingly, gene ontology (GO) analysis revealed that downregulated STAT5 target genes ($n = 476$) were principally enriched in the regulation of NF-κB activity, as well as other terms such as response to cytokines, transcription, and intracellular signal transduction (Fig. 5g). Particularly, CTCF depletion in BMDCs led to a decrease in expression of NF-κB pathway genes, such as *Trim25*, *Irak2*, and *Myd88*, due to the attenuation of the JAK2/STAT5 signaling pathway (Fig. 5h).

Chromatin binding of STAT5 at the promoter region of the *Trim25* gene was clearly detected in WT BMDCs but was abrogated by CTCF depletion (Fig. 5i, left panel). A recent report found that TRIM25, identified as an E3-ubiquitin ligase, positively regulates NF-κB signaling by promoting K63-linked ubiquitination of TRAF2 and by enhancing the interaction between TRAF2 and TAK1 or IKKB[39]. Interestingly, comparable enrichment of STAT5 between WT and KO BMDCs was demonstrated at a distal enhancer located 30 kb downstream from the *Trim25* promoter. However, chromatin interaction between the *Trim25* promoter and STAT5-binding distal enhancer was significantly decreased in KO BMDCs (Fig. 5i, left panel). As a result, close physical proximity between the promoter and STAT5-dependent enhancer was not secured; this might lead to decreased expression of the *Trim25* gene in KO BMDCs. Another example of the NF-κB pathway genes being downregulated by defective JAK2/STAT5 signaling is provided by *Irak2*, which is reported to lead to TRAF6 ubiquitination, an event critical for TLR-mediated NF-κB activation[40]. Chromatin binding of STAT5 was clearly detected at distal enhancers but not at the promoter of the *Irak2* gene in WT BMDCs. CTCF depletion in BMDCs decreased STAT5 occupancy at the distal enhancers located in the intron of the *Irak2* gene and weakened chromatin interactions between the *Irak2* gene promoter and the STAT5-dependent distal enhancers, potentially resulting in the downregulation of the *Irak2* gene (Fig. 5i, right panel). The decreased expression of *Trim25* and *Irak2* could be further validated by lower H3K4me3 levels at their promoters in the CTCF-deficient BMDCs than in WT BMDCs (Fig. 5i). Noteworthy, rescue of STAT5 activity in CTCF-deficient BMDCs did not increase the expression of *Trim25* and *Irak2* (Supplementary Fig. 7). These results suggested that downregulation of these STAT5 target genes in CTCF-deficient BMDCs was not simply due to attenuated STAT5 signaling, but also caused by disrupted looping between enhancers and their target genes.

### Defective NF-κB activation in CTCF-deficient BMDCs

TLR stimulation in DCs principally induces an activation of the NF-κB pathway through signaling cascades downstream of TLR4 which phosphorylates the inhibitor of κB (IκB) for proteasomal degradation and subsequently releases NF-κB transcription factors[41,42]. As CTCF-deficient BMDCs demonstrated a diminished expression of NF-κB signaling components, we next examined whether the TLR4/NF-κB signaling pathways were dysregulated by CTCF depletion. Phosphorylation of IκBα after LPS stimulation was dramatically induced in WT BMDCs but not in KO BMDCs, while the expression of the RelA subunit of the NF-κB complex was not changed by CTCF depletion (Fig. 6a and Supplementary Fig. 8a). In addition, immunofluorescence and western blot analysis demonstrated that the nuclear translocation of RelA following LPS stimulation was considerably reduced by CTCF depletion (Fig. 6b–d, and Supplementary Fig. 8b). Furthermore, ChIP-seq analysis revealed that LPS stimulation potently induced genome-wide chromatin binding of RelA in WT BMDCs but not in KO BMDCs (Fig. 6e–g). These results indicated that NF-κB activation following TLR4 stimulation is severely compromised in CTCF-deficient BMDCs.

Notably, the chromatin binding of RelA was accompanied by the induced enrichments of SMC1 and H3K27ac following LPS stimulation in WT BMDCs but not in KO BMDCs (Fig. 6h, i). Given that SMC1 and H3K27ac are known to be tightly associated with loop formation and enhancer activation, respectively, these results suggested that dynamic NF-κB binding may contribute to the changes in loop formation and enhancer activity. Indeed, LOLA enrichment analysis of the H3K27ac HiChIP loops in untreated and LPS-stimulated WT BMDCs (WT vs. WTL) demonstrated that sites bound by RelA were strongly represented at the anchors associated with "gain of loop to activation-inducible enhancer by LPS stimulation" (Fig. 6j). Interestingly, RelA binding was also significantly represented at the anchors associated with "weaker loop from less active enhancer by CTCF depletion", when H3K27 HiChIP loops were compared between WT and KO BMDCs following LPS stimulation (WTL vs. KOL) (Fig. 6k). Furthermore, the number of H3K27ac HiChIP loops overlapped with RelA ChIP-seq peaks at least at one of the loop anchors was dramatically increased by LPS stimulation in WT BMDCs but not in KO BMDCs (Fig. 6l). These results clearly suggested that LPS stimulation induces NF-κB binding which may facilitate stronger enhancer-promoter interactions as well as higher enhancer activity in WT BMDCs but not in CTCF-deficient BMDCs.

### RelA-mediated long-range chromatin interactions in LPS-stimulated BMDCs

To further gain insight into the impact of NF-κB activation on the higher order genome organization, we performed RelA HiChIP for the LPS-stimulated WT BMDCs and identified ~20,000 statistically significant RelA-mediated long-range chromatin interactions using FitHiChIP at a resolution of 10 kb with an FDR < 10⁻² (Fig. 7a). The majority (~40%) of RelA HiChIP loops occurred between enhancers, whereas about 20% occurred between promoters and enhancers (Supplementary Fig. 9a). Given that more than 76% (4153/5465) of RelA binding sites in LPS-stimulated WT BMDCs were located distally from promoters (Supplementary Fig. 9b), we utilized RelA HiChIP loops to

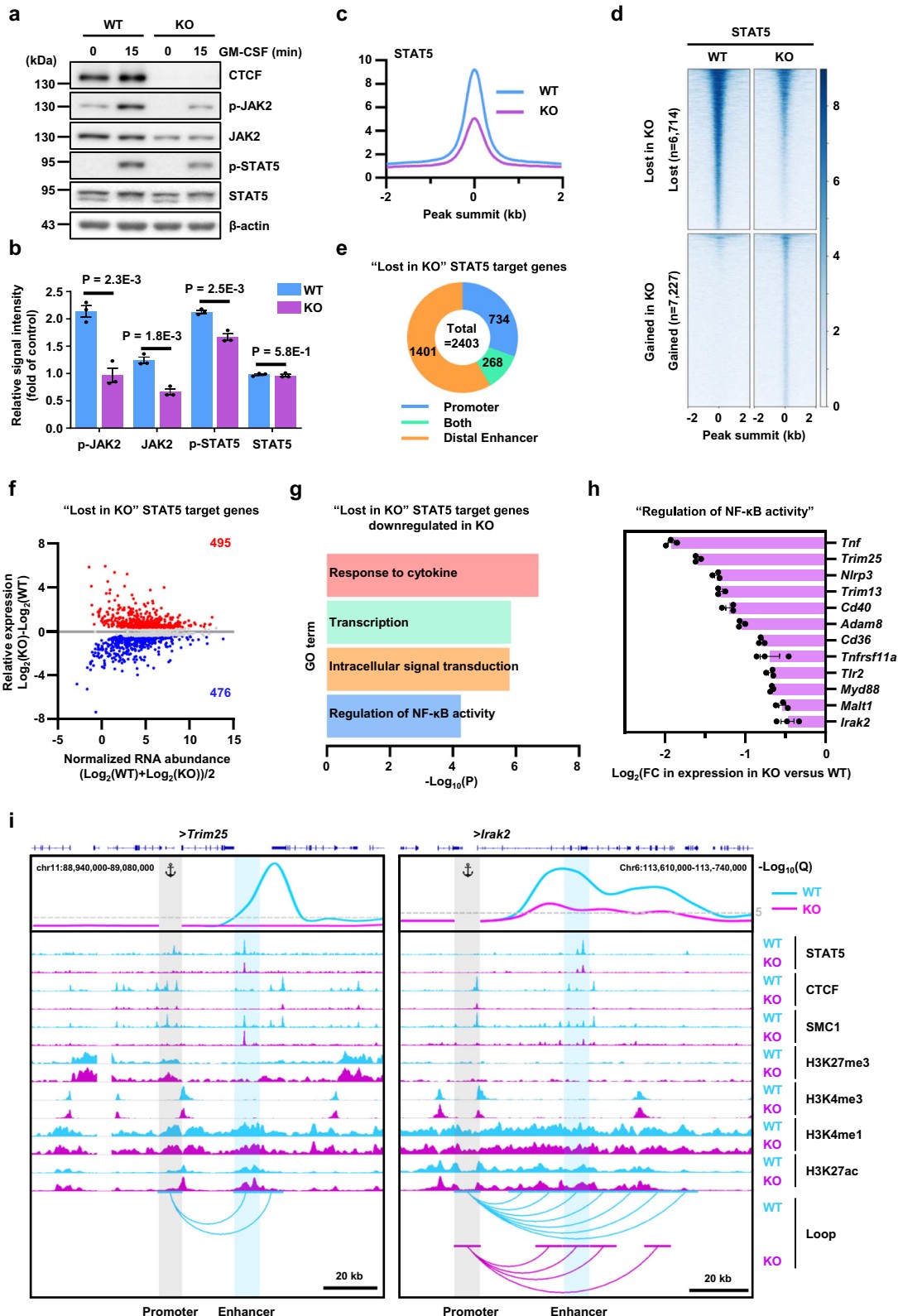

map direct RelA target genes. While 846 (499 plus 347) genes were identified by overlapping promoters with RelA ChIP-seq peaks, a much greater number of direct RelA target genes (1539) were found where RelA binds to distal enhancers but not to promoters (Supplementary Fig. 9c). The example of three different types of RelA target genes classified by the overlap feature of RelA peaks with promoters and distal enhancers are provided in Supplementary Fig. 9d. While binding

of RelA to the promoters was observed at both the *Il1f6* and *Acod1* genes, only the latter gene showed additional regulation by RelA by forming loops between its promoter and upstream enhancer regions (Supplementary Fig. 9d, left and middle panels). Furthermore, we could identify the *Ch25h* gene as the direct RelA target gene, even though very little RelA binding was observed at the promoter region of this gene, since several distal enhancer regions with RelA binding

**Fig. 5 | Defective JAK2/STAT5 signaling in CTCF-deficient BMDCs leads to downregulation of NF-κB pathway component genes. a** Western blotting performed with the indicated antibodies. The data are representative of three independent experiments with similar results. **b** Relative signal intensities of proteins in **a** were measured using ImageJ software. Error bars represent mean ± standard error of the mean (s.e.m.). Significance was calculated using an unpaired two-tailed *t*-test using *n* = 3 independent samples. **c** Histogram showing the average tag density of STAT5 ChIP-seq peaks called for WT and KO BMDCs. **d** Heatmap of ChIP-seq signals with ±2 kb of unique STAT5 peaks comparing STAT5 enrichment (fold change > 1.3; FDR < 0.05) between WT and KO BMDCs. The data are representative of two independent experiments with similar results. **e** A pie chart depicting the numbers of genes, assigned to "lost in KO" STAT5 peaks and categorized by the overlap feature of the STAT5 peaks with promoters, distal enhancers, or both. **f** RNA-seq MA plot for the genes assigned to "lost in KO" STAT5 peaks. The number of genes

exhibiting >1.3-fold decreases in WT (red) or KO (blue) BMDCs with a false discovery rate <0.05 has been indicated. **g** Enrichment of pathway terms on the "Lost in KO" STAT5 target genes whose RNA expression was downregulated in KO. Significance was calculated by one-sided Fisher's Exact test. **h** Changes in RNA expression of the genes associated with the pathway term of "Regulation of NF-κB activity". Error bars represent mean ± standard error of the mean (s.e.m.). *n* = 3 biologically independent samples. **i** Snapshots displaying virtual 4C plots, ChIP-seq signal tracks, and significant loops (from top to bottom) at the *Trim25* (left) and *Irak2* (right) loci. Virtual 4C plots (V4C) shows normalized H3K27ac HiChIP loop strength (represented as -Log10(Q)) with the TSSs of *Trim25* (left) and *Irak2* (right) genes as the viewpoint. IGV browser shows ChIP-seq signal tracks for STAT5, CTCF, SMC1, H3K27me3, H3K4me3, H3K4me1, and H3K27ac. Arcs show significant interactions with −Log10(Q) ≥ 5. Only the loops interacting with the viewpoint have been displayed. Source data are provided as a Source Data file.

simultaneously interacted with the *Ch25h* promoter (Supplementary Fig. 9d, right panels).

## Compromised expression of RelA-dependent pro-inflammatory cytokines in CTCF-deficient BMDCs

We then analyzed RNA-seq data to explore whether LPS-induced gene expression was affected by CTCF depletion. Out of 1579 upregulated genes by LPS stimulation in WT BMDCs (WTL vs. WT) (Fig. 3d), 403 were significantly downregulated by CTCF depletion after LPS stimulation (KOL vs. WTL) (Supplementary Fig. 9e). Gene ontology analysis showed that these 403 genes were enriched in positive regulation of cytokine production and inflammatory-related pathways (Supplementary Fig. 9f).

Given that LPS-induced NF-κB activation was defective in CTCF-deficient BMDCs, we next analyzed transcriptomic data and identified the RelA target genes which exhibited upregulation by LPS stimulation (WT vs. WTL) but downregulation by CTCF depletion (WTL vs. KOL). Gene ontology (GO) analysis showed that these CTCF-dependent RelA target genes (*n* = 169, lower right of Fig. 7b) were enriched in inflammation-related pathways, including cytokine production, nitric oxide (NO) biosynthetic process, inflammatory response, and chemotaxis (Fig. 7c). Particularly, CTCF-deficient BMDCs failed to produce pro-inflammatory cytokines, including *Il6*, *Il12a*, and *Il12b*, when activated by LPS stimulation (Fig. 7d).

Activation of WT BMDCs by LPS treatment led to a dramatic increase of the RelA binding at the *Il6* promoter as well as at the upstream SE region spanning ~200 kb, where enrichment of SMC1 and H3K27ac were also upregulated (Fig. 7e). V4C plot for H3K27ac HiChIP loop with the *Il6* promoter region as the anchor clearly demonstrated that LPS stimulation leads to a dramatic increase in the loop strength between the *Il6* promoter with multiple RelA binding sites located at the SE (Fig. 7e). In contrast to the *Il6* gene, the role of RelA in the LPS-induced expression of the *Il12a* gene was mediated mostly by distal enhancers but not by a promoter, since we could observe very little RelA binding to the *Il12a* promoter in LPS-stimulated WT BMDCs (Fig. 7f). RelA binding to the distal enhancers following LPS stimulation was accompanied by the establishment of an active SE, which showed a dramatic increase in the loop formation with the *Il12a* promoter (Fig. 7f). RelA HiChIP loops in the *Il6* or *Il12a* loci could validate the important role of NF-κB in the establishment of chromatin interactions between multiple regulatory elements within SEs as well as between SE and promoters (Figs. 7e, f). The NF-κB-centric regulatory loops required for the upregulation of the *Il6* and *Il12a* genes, however, were abrogated by CTCF depletion, given that LPS stimulation of CTCF-deficient BMDCs displayed very little RelA binding, aborted activation of SEs, and very weak interaction between the SE and promoter (Fig. 7e, f). The 3D clique analysis of the *Il6* and *Il12a* loci also demonstrated that LPS stimulation drives the formation of the highly interacting spatial clusters between activation-inducible enhancers and promoters in WT BMDCs but not in CTCF-deficient BMDCs (Fig. 7g, h). The results clearly indicate that

CTCF is essential in establishing the NF-κB-dependent 3D enhancer networks required for optimal expression of pro-inflammatory cytokine genes such as *Il6* and *Il12a*.

To further validate the role of NF-κB activation in facilitating enhancer activation and loop formation, we blocked the NF-κB signals using JSH-23, which prevents the nuclear translocation of NF-κB RelA[43] while leaving CTCF intact (Fig. 8a, b). First, H3K27ac ChIP-seq analysis demonstrated that enrichment of H3K27ac at RelA binding sites induced by LPS stimulation was almost completely abrogated by JSH-23 pre-treatment (Fig. 8c). Next, the RelA-mediated chromatin interactions (shown in Fig. 7a) were subject to differential H3K27ac HiChIP loop analysis, which revealed 346 gained loops and 295 lost loops due to LPS stimulation (Supplementary Fig. 10a). Interestingly, these dynamic changes in RelA-mediated chromatin interactions induced by LPS stimulation were significantly attenuated by pre-treatment of the BMDCs with JSH-23 (Fig. 8d). Further, RelA target genes (shown in Supplementary Fig. 9c) were analyzed by RNA-seq, which revealed 853 upregulated and 407 downregulated genes in response to LPS stimulation (Supplementary Fig. 10b). The extent of changes in the transcript abundance of these differentially expressed RelA target genes was significantly diminished by pre-treatment of the BMDCs with JSH-23 before LPS stimulation (Fig. 8e). The best examples showing the effect of NF-κB inhibition on enhancer activation, loop formation, and the resultant RNA expression profile were provided by *Il6* and *Il12a* (Fig. 8f). LPS stimulation resulted in increased mRNA expression accompanied by enhanced H3K27ac levels upstream of *Il6* and *Il12a* promoters, which were severely abrogated by pre-treatment of the BMDCs with JSH-23 (Fig. 8f). Moreover, V4C plot for H3K27ac HiChIP loop with the *Il6* and *Il12a* promoter regions as anchor demonstrated that loop formation induced by LPS stimulation between promoters and RelA binding sites were also decreased by JSH-23 pre-treatment (Fig. 8f). Taken together, these results indicated that NF-κB signaling can control spatial enhancer-promoter proximity, as well as distal enhancer activities for the optimal expression of its target genes.

## Differentiation of Th1 and Th17 cells is attenuated in CTCF-deficient DCs

Impaired production of IL-6 and IL-12 in CTCF-deficient BMDCs was further validated at both the mRNA and protein levels (Fig. 9a, b). Given that IL-12 and IL-6 are well-known cytokines capable of driving Th1 and Th17 differentiation, respectively, we examined the capacity of CTCF-deficient BMDCs for T cell differentiation. When naïve CD4[+] T cells were cultured with BMDCs under the Th1- and Th17-driving conditions, CTCF-deficient BMDCs showed significantly lower activities to mediate expression of IFN-γ and IL-17A, respectively, in CD4[+] T cells (Fig. 9c, d). The reduced T cell differentiation was largely attributed to the defective cytokine production in the CTCF-deficient BMDCs; treatment of exogenous IL-12 and IL-6 significantly rescued Th1 and Th17 differentiation by CTCF-deficient BMDCs, respectively (Fig. 9c, d). These data indicate that CTCF is essential for the maximal

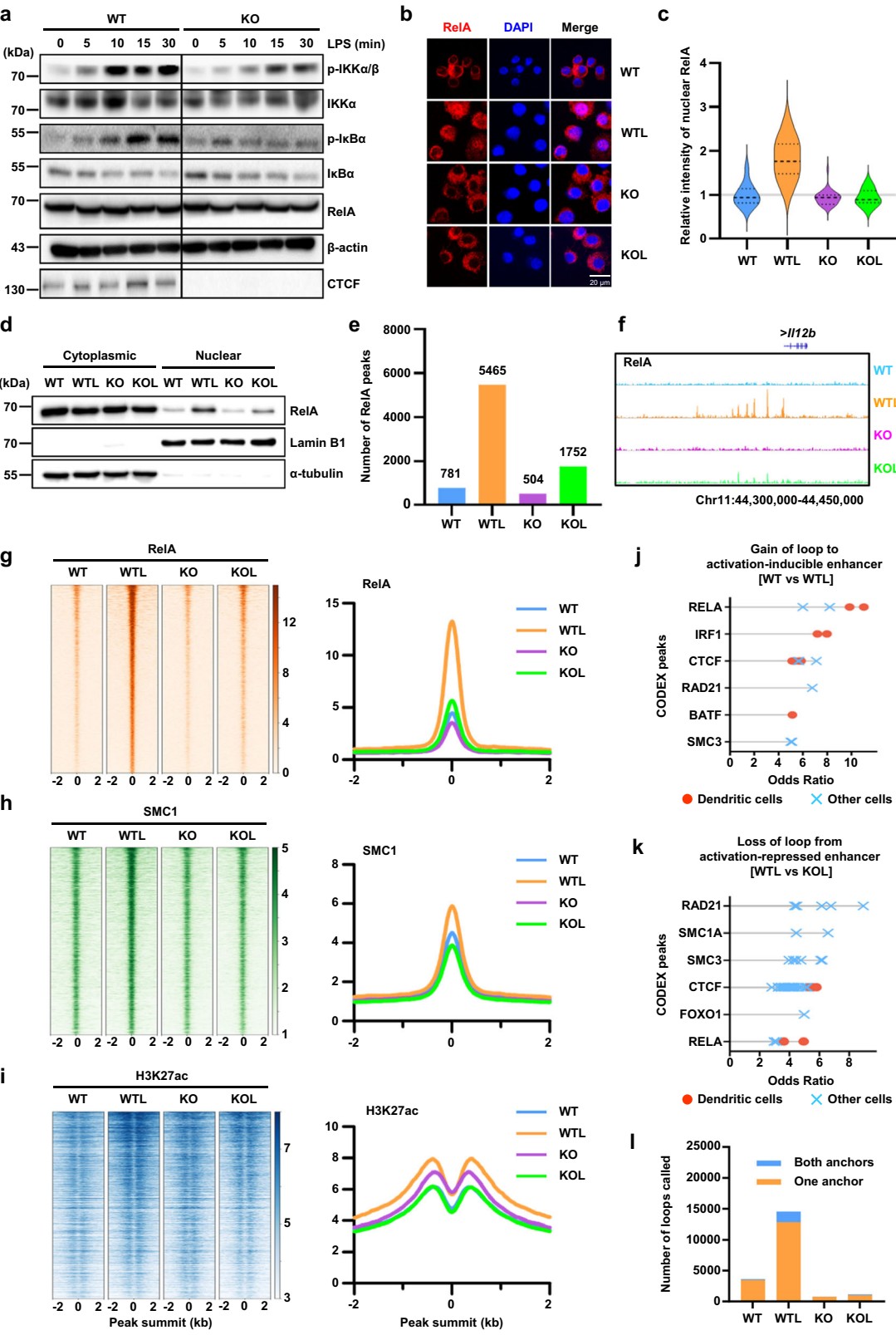

expression of IL-12 and IL-6 from BMDCs to prime Th1 and Th17 cell differentiation, respectively.

## Discussion

In this study, we performed an integrated analysis of the three-dimensional genome organization, genome-wide enhancer profile, and their relationship with gene expression. The results revealed that both enhancer-promoter communication and the enhancer landscape are dynamically reprogrammed when BMDCs establish a pro-inflammatory transcriptional program in response to pathogenic stimuli, although LPS stimulation did not induce changes in global genome organization at the compartment and TAD levels. Interestingly, the stimulus-induced enhancer activation can be instructive for specific enhancer-promoter interactions, given that spatial proximity for

**Fig. 6 | Attenuation of LPS-induced NF-κB activation in CTCF-deficient BMDCs.** **a** WT and KO BMDCs were stimulated with LPS and western blotting was performed with indicated antibodies. The data are representative of three independent experiments with similar results. **b** Representative immunofluorescence staining for RelA (red) and DAPI (blue). The data are representative of three independent experiments with similar results. **c** Relative fluorescence intensities of nuclear RelA in **b** was measured using ImageJ software. **d** Western blot analysis of LPS-induced nuclear translocation of RelA in WT and KO BMDCs. The data are representative of three independent experiments with similar results. **e** Bar graph showing the number of RelA ChIP-seq peaks called. **f** Genomic snapshot of RelA ChIP-seq at the *Il12b* locus. **g**–**i** Heatmaps of normalized ChIP-Seq tag densities for RelA (**g**), SMC1 (**h**), and H3K27ac (**i**) at RelA binding sites called from LPS-stimulated wild-type

BMDCs (WTL). Histogram showing the average tag density of ChIP-seq peaks were displayed on right of the heatmaps. The data are representative of two independent experiments. **j** Relative LOLA enrichment of transcription factors at the anchors of H3K27ac HiChIP loops showing regulatory mode of "gain of loop to newly activated enhancer" when WTL was compared to WT. **k** Relative LOLA enrichment of transcription factors at the anchors of H3K27ac HiChIP loops showing regulatory mode of "loss of loop to newly repressed enhancer" when KOL was compared to WTL. Each dot represents a TF ChIP-seq dataset from dendritic cells, and ChIP-seq datasets from other cell types are shown as crosses (**j**, **k**). **l** The number of H3K27ac HiChIP loops overlapped with RelA ChIP-seq peaks at either one or both loop anchors. Source data are provided as a Source Data file.

some enhancers with their target promoters were increased as the activities of enhancers were strengthened by LPS stimulation. In contrast, weakened enhancer activities following BMDC activation can lead to diminished chromatin interactions with their target promoters. Alteration in both enhancer activities and enhancer-promoter interactions exhibited positive correlations with changes in the transcriptional activity of the gene they control. However, formation or disruption of enhancer-promoter interactions can alter transcriptional activity, even without concomitant changes in the chromatin state of enhancers during BMDC activation. The results indicate that transcriptional regulation of a significant number of BMDC genes depends on the rewiring of chromatin interactions between constitutively active enhancers and their target genes following LPS stimulation. In contrast, spatial proximity between enhancer and promoter does not automatically induce gene activation, given that quite a few enhancer-promoter interactions seem to be established before LPS stimulation. Such pre-formed enhancer-promoter contacts may 'prime' some genes for transcriptional activation, until an additional trigger mediated by LPS stimulation boosts transcription for maximal gene expression. Taken together, the results indicate that LPS stimulation lead to dramatic changes in spatial enhancer-promoter proximity and chromatin state of distal enhancers, both of which contribute to the diverse mode of regulatory mechanisms for the accurate control of gene expression during BMDC activation.

As reported in other cellular models[30–32,44], CTCF depletion in BMDCs does not perturb the A/B compartment, but disrupts TAD boundary integrity. Moreover, we observed that enhancer-centric chromatin interactions were significantly decreased in CTCF-deficient BMDCs, while much weaker insulation capacity caused by CTCF depletion led to aberrant enhancer-promoter interactions and dysregulated RNA expression. Remarkably, CTCF-deficient BMDCs demonstrated a profound defect in producing pro-inflammatory cytokines such as IL-6 and IL-12 in response to LPS, suggesting that CTCF is essential for the terminal differentiation of BMDCs and their effective immune responses. GM-CSF depends on the JAK2/STAT5 signaling pathway to efficiently promote DC differentiation from hematopoietic progenitor cells[38,45,46]. Interestingly, we could reveal that JAK2/STAT5 signaling was attenuated by CTCF depletion during the GM-CSF-mediated DC differentiation, since our western blot and ChIP-seq analysis demonstrated that phosphorylation of JAK2 and STAT5 by GM-CSF as well as genome-wide STAT5 occupancy was significantly decreased in CTCF-deficient BMDCs. Moreover, combined analysis of H3K27ac HiChIP, STAT5 ChIP-seq, and RNA-seq allowed us to identify the STAT5 direct target genes whose expressions were affected by attenuated JAK2/STAT5 signaling. Surprisingly, the expression of various signaling molecules involved in TLR4 downstream signaling cascades was commonly downregulated in CTCF-deficient BMDCs, thereby having an attenuated NF-κB activation and defective pro-inflammatory cytokine production in response to LPS stimulation. These results demonstrated that the ability of BMDCs to respond to TLR4 signaling cues is hardwired into the 3D genome architecture

during GM-CSF-mediated differentiation, which permits a rapid and accurate NF-κB-centric transcriptional response.

The transcription factor NF-κB regulates multiple aspects of TLR-mediated activation of DCs and serves as a vital mediator of inflammatory responses[41,42]. The molecular mechanism through which NF-κB regulates the inducible expression of inflammatory genes has been extensively studied[41,42]. Integrated analysis with genome-wide binding of multiple TFs and histone modifications revealed that dynamic alterations of the H3K9K14ac levels following LPS stimulation were tightly linked with stimulus-induced binding of NF-κB at inducible promoters and their enhancers[17]. Moreover, emerging works demonstrated that the binding of NF-κB to regulatory elements can induce changes in enhancer-promoter interactions in response to inflammatory stimuli[47]. For example, binding of NF-κB to a distal enhancer facilitated chromatin looping in the OPN promoter to drive LPS-stimulated OPN expression in murine macrophages[47]. Consistent with these reports, we could observe that augmented binding of RelA induced by LPS stimulation plays a crucial role in establishing elevated enhancer activities as well as favorable 3D chromatin structure for maximal expression of target genes. Among the various pro-inflammatory cytokines and chemokines induced by NF-κB signaling, IL-12 and IL-6 are known to be the key drivers for the differentiation of naïve CD4+ T lymphocytes to IFN-γ-producing Th1 and IL-17-producing Th17 effectors, respectively[48]. Interestingly, our H3K27ac HiChIP analysis revealed that stimulus-induced binding of NF-κB to SEs located far from promoters facilitated the formation of highly interacting enhancer-promoter spatial clusters in the *Il6* and *Il12a* genes. However, such hyperconnected 3D cliques were abrogated in CTCF-depleted BMDCs possibly due to defective NF-κB signaling as well as weakened NF-κB-mediated chromatin looping, which then resulted in compromised expression of RelA-dependent pro-inflammatory cytokines and attenuated differentiation of Th1 and Th17 cells. These results suggested that non-permissive chromatin topology that prevents close spatial proximity between distal enhancer regions and target genes can function as a regulatory barrier that must be resolved for proper gene expression.

In conclusion, CTCF is essential in establishing the NF-κB-dependent three-dimensional enhancer network underlying inflammatory responses of BMDC, and dysregulated CTCF can contribute to the pathogenesis of a number of inflammatory diseases.

## Methods
### Mice and cells
All mouse experimental procedures were approved by the Department of Laboratory Animal Resources Committee of Yonsei University College of Medicine. C57BL/6 mice carrying a conditional Ctcf allele (Ctcf[fl/fl]) were crossed with Rosa26-CreER (CreER) C57BL/6 mice to generate a tamoxifen-inducible Ctcf conditional knockout strain (CreER;CTCF[fl/fl]) as described in our previous study[24]. Male and female 8-12 weeks old mice that were bred in specific pathogen-free facilities at Yonsei University College of Medicine were used for all experiments with 12-hour light/dark cycles. Room temperature was

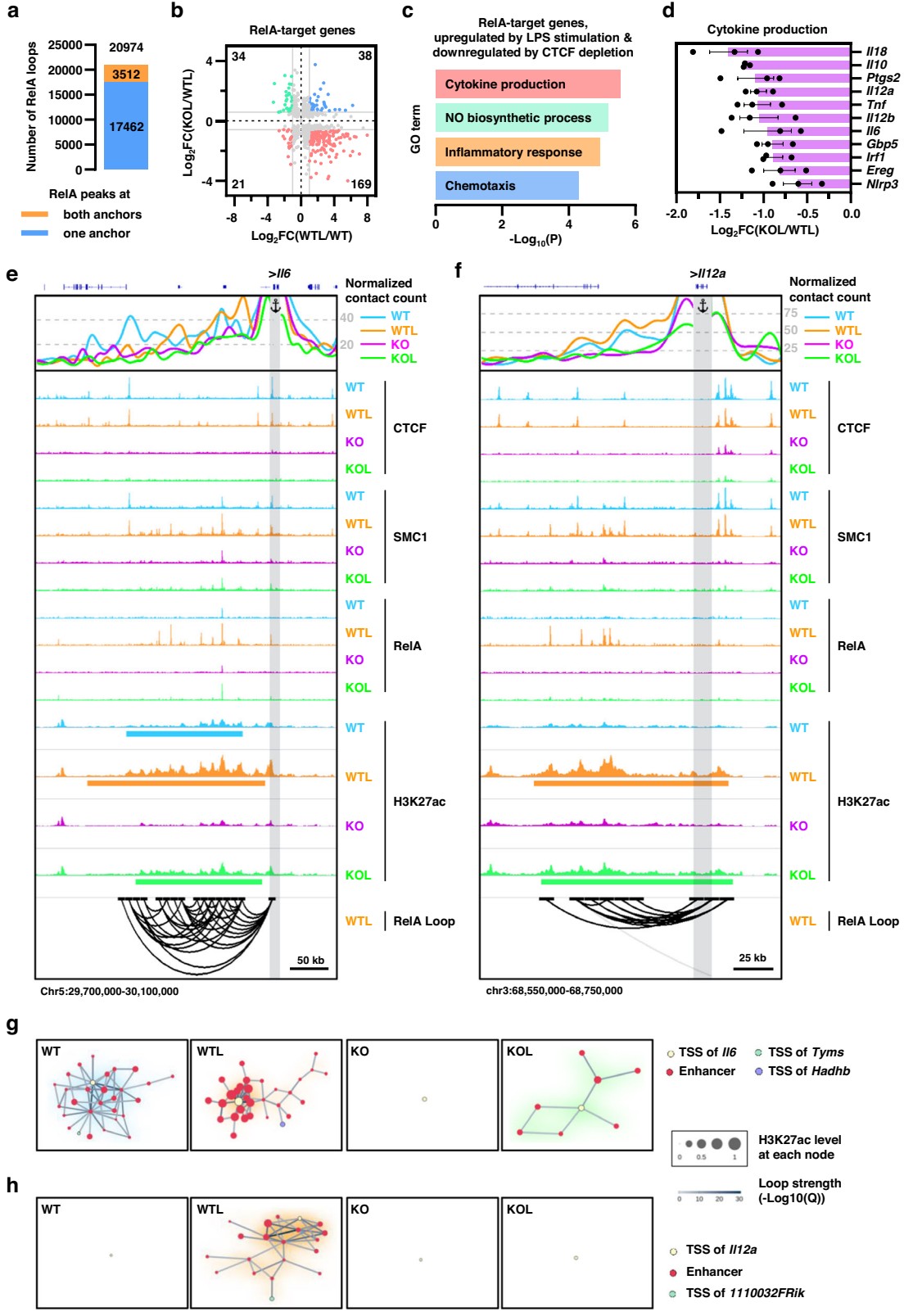

maintained at $23 \pm 1\,^{\circ}\mathrm{C}$ and humidity level was controlled between
40–60%. Age- and sex-matched CreER littermate mice were used as
wild-type (WT) controls throughout the study. BM cells prepared
from CreER;CTCF[wt/wt] or CreER;CTCF[fl/fl] mice were cultured in RPMI
1640 (HyClone, SH30027.01) supplemented with 10% fetal bovine
serum (HyClone, SH30071.03), 100 U/mL penicillin, 100 µg/mL
streptomycin (HyClone, SV30010), and 50 µM 2-mercaptoethanol

(Gibco, 21985-023) in the presence of murine granulocyte/monocyte
colony-stimulating factor (GM-CSF, provided by Chae Gyu Park,
Yonsei University College of Medicine) for 6 days. The medium was
changed every 2 days. For the deletion of the Ctcf allele in vitro,
4-OH-tamoxifen (Sigma-Aldrich, H7904) dissolved in 100% ethanol
was added on the first day of culture (final 0.5 µM). Floating
cells were harvested and CD11c[+] cells were positively selected as

**Fig. 7 | LPS-stimulated expression of pro-inflammatory cytokines was diminished in CTCF-deficient BMDCs. a** Number of the high-confidence RelA-mediated chromatin interactions in WTL. The number of RelA HiChIP loops overlapped with RelA ChIP-seq peaks at either one or both anchors was indicated in the bar. **b** Scatter plot showing Log2-fold changes in RNA expression levels of RelA direct target genes shown in Supplementary Fig. 9c. The number of genes exhibiting >2-fold changes in WTL versus WT and 1.5-fold changes in KOL versus WTL with a false discovery rate <0.05 is indicated at each quartile. **c** Bar plot of −Log10 *P*-value showing enrichment of gene ontology terms (biological process) associated with RelA direct target genes whose expressions were upregulated by LPS stimulation and downregulated by CTCF depletion. Significance was calculated by one-sided Fisher's Exact test. **d** Changes in RNA expression of the genes associated with the pathway term of "cytokine production". Error bars represent mean ± standard error of the mean (s.e.m.). *n* = 3 biologically independent samples. **e, f** Snapshot showing virtual 4C plots, ChIP-seq signal tracks, and significant RelA loops (from top to

bottom) at the *Il6* (**e**) and *Il12a* (**f**) loci. Sky blue, orange, purple, and green represent unstimulated wild-type BMDCs (WT), LPS-stimulated wild-type BMDCs (WTL), unstimulated CTCF knockout BMDCs (KO), and LPS-stimulated CTCF knockout BMDCs (KOL), respectively. Virtual 4C plots (V4C) shows normalized H3K27ac HiChIP loop strength (represented as −Log10(*Q*)) with the TSSs of *Il6* and *Il12a* genes as the viewpoint. IGV browser shows ChIP-seq signal tracks for CTCF, SMC1, RelA, H3K27ac. The location of super-enhancers (SE), if any, were shown under H3K27ac ChIP-seq signal tracks. RelA loops shows significant interactions with −Log10(*Q*) ≥ 2. Gray vertical bars highlight the location of viewpoints. **g, h** 3D cliques for *Il6* (**g**) and *Il12a* (**h**) loci where each node represented a promoter or an enhancer and each edge represented a significant chromatin interaction of H3K27ac HiChIP loop with −Log10(*Q*) ≥ 5. The size of each node indicates the H3K27ac levels and the color of each edge indicates the loop strength (-Log10(*Q*)). Source data are provided as a Source Data file.

BM-derived dendritic cells (BMDCs) using the CD11c microbead (Miltenyi Biotec, 130-125-835). The cells were stimulated with LPS (Invivogen, E. coli K12 strain) at a concentration of 100 ng/mL for the indicated times.

### RNA extraction and quantitative real-time polymerase chain reaction (RT-qPCR)

One million BMDCs were stimulated with LPS for 3 h and subjected to RNA extraction using Hybrid-R Total RNA kit (GeneAll Biotechnology, 305-101) according to the manufacturer's instruction. RNAs were reverse transcribed using PrimeScript™ RT Master Mix (Takara Bio, RR036A). The resulting cDNAs were subjected to qRT-PCR using the ABI StepOnePlus real-time PCR system (Applied Biosystems) and QuantStudio3 (Applied Biosystems); the synthesis of double-stranded DNA was monitored during various PCR cycles using QuantiNova SYBR Green PCR Kit (Qiagen, 208052). For each sample, duplicate test reactions were analyzed for the expression of the gene of interest, and the results were normalized to Rpl7 mRNA levels. The primer sequences are listed in Supplementary Table 1. Three biological replicates were performed for each condition.

### Western blotting

BMDCs were lysed in T-PER™ Tissue Protein Extraction Reagent (Thermo Fisher Scientific, 78510) with a protease and phosphate inhibitor cocktail (Thermo Fisher Scientific, 78440). Proteins were separated using sodium dodecyl sulfate-polyacrylamide gel electrophoresis and transferred onto a polyvinylidene fluoride membrane. After blocking with 5% skim milk, the membrane was incubated with primary antibodies with 1:1000 dilution: β-actin (sc-47778) and α-tubulin (sc32293) from Santa Cruz; CTCF (2899) p-JAK2 (3771), JAK2 (3230) p-STAT5 (9351), STAT5 (94205), p-IKKα/β (2078), IKKα (2682), p-IκBα (2859), IκBα (9242), and RelA (8242) from Cell Signaling Technology; ALDH1A2 (ab156019) and Lamin B1 (ab133741) from Abcam, followed by incubation with horseradish peroxidase-conjugated secondary antibody with 1:2000 dilution: HPR-linked anti-Rabbit IgG (7074), and HPR-linked anti-Mouse IgG (7076) from Cell Signaling Technology. The target proteins were visualized using Pierce Western Blotting Substrate (Thermo Fisher Scientific, 32132) and Image Quant LAS 4000 (GE Healthcare). Three biological replicates were performed for each condition.

### Flow cytometry

Fluorochrome-conjugated anti-mouse CD4 (RM4-5), CD8a (53-6.7), CD11b (M1/70), CD11c (N418), CD45.1 (A20), CD45.2 (104), CD80 (16-10A1), CD86 (GL1), I-A/E (M5/114.15.2), IFN-γ (XMG1.2), and IL-17A (eBio17B7), and CD135 (A2F10) were obtained from eBioscience and CD115 (T38-320) was obtained from BD Biosciences. All antibody dilutions were 1:200. Cell death and apoptosis were analyzed using an Annexin V/Propidium iodide (PI) staining kit (eBioscience, 88-8007-74).

according to the manufacturer's protocol. Cell proliferation was determined using CFSE staining (Molecular Probes, C1157). Aldehyde dehydrogenase (ALDH) activity was measured in the BMDCs using an ALDEFLUOR Kit (STEMCELL Technologies, 01700) according to the manufacturer's protocol. For the intracellular detection of IFN-γ and IL-17A, cells were fixed and permeabilized with Foxp3/Transcription Factor Staining Buffer Set (eBioscience, 00-5523-00) and incubated for 30 min on ice with the relevant antibodies diluted in Permeabilization buffer (eBioscience). Cytokines secreted into culture media were quantified using the CBA Mouse Inflammation Kit (BD Biosciences, 552364) and the CBA Mouse Th1/Th2/Th17 Cytokine Kit (BD Biosciences, 560485). Stained cells or samples were analyzed using flow cytometry with the FACSVerse system (BD Biosciences) and a FACS LSR Fortessa flow cytometer (BD Biosciences). All flow cytometry data collected by FACSuite (BD Biosciences) and FACSDiva (BD Biosciences) were analyzed using the FlowJo software (Treestar). CBA data were analyzed using FCAP Array software (BD Biosciences). Three biological replicates were performed for each condition.

### Antigen uptake assay

BMDCs were washed with 1× PBS and incubated on ice for 1 min to maintain normal cell conditions. Subsequently, the cells were incubated with or without the Alexa-647 conjugated OVA protein (Invitrogen, O34784) at 4 °C or 37 °C for 30 min. Stained cells were analyzed using the FACSVerse system (BD Biosciences). Three biological replicates were performed for each condition.

### T cell proliferation assay

CD4+ T cells were isolated from spleen of OTII mouse and purified using the MagniSort Mouse CD4 positive selection Kit (eBioscience, 8802-6841-74). Purified T cells were stained with CFSE and co-cultured with OVA323-339 peptide (Sigma, O1641) pre-educated CD11c+ BMDCs for 3 days. Three biological replicates were performed for each condition.

### Helper T cell differentiation in vitro

CD4+ T cell in vitro differentiation with BMDCs was performed as previously described[49]. In brief, CD4+ naïve T cells were isolated from the spleen of C57BL6 mice using the MagniSort Mouse CD4+ Naïve T cell Enrichment Kit (eBioscience, 8804-6824-74). Naive CD4+ T cells (1 × 10^5 cells/well) and CD11c+ BMDCs (1 × 10^4 cells/well) were co-cultured in the presence of soluble anti-CD3ε (0.15 μg/mL; BioLegend, 100331) for 4 days. For Th1 differentiation, BMDCs were treated with LPS (100 ng/mL). For Th17 differentiation, BMDCs were stimulated with LPS (100 ng/mL) and TGF-β (3 ng/mL; Peprotech, 100-21 C). In vitro differentiated CD4+ T cells were incubated with 50 μg/mL PMA (Sigma, P1585) and 1 μg/mL ionomycin (Sigma, I3909) in the presence of brefeldin A (eBioscience, B6542) for 4 h before intracellular cytokine staining analysis. Three biological replicates were performed for each condition.

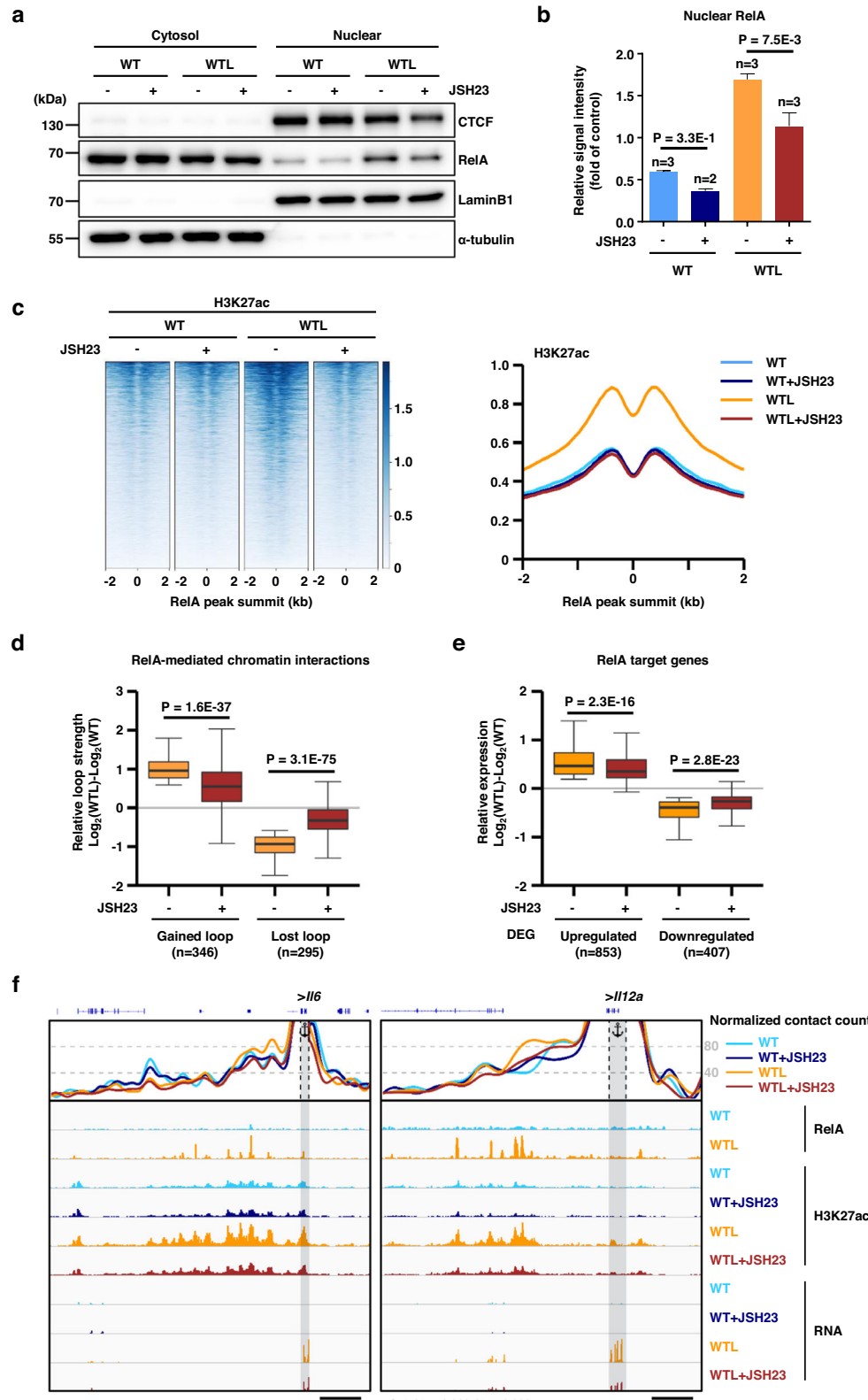

### Immunofluorescence assay

After LPS stimulation, BMDCs were fixed with methanol for 10 min at −20 °C, and permeabilized with 0.5% Triton X-100 for 10 min at RT on slide glass. Cells were incubated overnight at 4 °C with RelA primary antibody (Cell Signaling Technology, 8242) with 1:100 dilution followed by 2 h incubation at RT with Alexa Fluor 594 goat anti-rabbit IgG secondary antibody (Invitrogen,

A-11037). DNA was stained with DAPI included in the mounting medium (VECTOR Laboratories, H-1200). Images were acquired using a LSM 700 confocal microscope (Zeiss) and processed with ZEN software (Zeiss). Fluorescence intensity of nucleus and cytoplasm was analyzed using the ImageJ software (National Institutes of Health). Three biological replicates were performed for each condition.

**Fig. 8 | NF-κB signaling controls enhancer activation and enhancer-promoter interactions for optimal expression of its target genes. a** WT BMDCs were either untreated or treated with JSH-23 for 3 h and Western blot analysis was performed to assess LPS-induced RelA nuclear translocation. The data are representative of three independent experiments with similar results. **b** Densitometric analysis of the proteins in **a** was performed using ImageJ. Error bars represent mean ± standard error of the mean (s.e.m.). Significance was calculated using a two-way ANOVA with multiple comparisons of Bonferroni post-test. The sample sizes (n) were labeled in the figure. **c** Heatmaps of normalized ChIP-seq tag densities for H3K27ac at RelA binding sites called from LPS-stimulated WT BMDCs (WTL). Histogram showing the average tag density of ChIP-seq peaks are displayed at the right of the heatmaps. The data are representative of two independent experiments. **d** The effect of JSH-23 on the extent of changes in the chromatin interactions of either gained or lost RelA loops in response to LPS stimulation was analyzed by H3K27ac HiChIP. **e** The effect

of JSH-23 on the transcript abundance of either upregulated or downregulated RelA target genes in response to LPS stimulation was analyzed by RNA-seq. DEG: differentially expressed gene. Box-plot in **d**, **e** with midline = median, box limits = Q1 (25th percentile)/Q3 (75th percentile), whiskers = minimum and maximum values. Significance in **d**, **e** was calculated using two-sided Wilcoxon rank sum test. The sample sizes (n) were labeled in the figure. **f** Snapshot showing virtual 4C plots (V4C), ChIP-seq signal tracks, and RNA-seq signal tracks (from top to bottom) at the *Il6* (left) and *Il12a* (right) loci. Sky blue, blue, orange, and brown represent unstimulated WT BMDCs in the absence of JSH-23 (WT) or pretreated with JSH-23 (WT + JSH23), LPS-stimulated WT BMDCs in the absence of JSH-23 (WTL) or pretreated with JSH-23 (WTL + JSH23), respectively. V4C shows normalized H3K27ac HiChIP contact counts with the TSSs of *Il6* and *Il12a* as the viewpoint. Source data are provided as a Source Data file.

## Retroviral constructs and virus production

Constitutively active STAT5a mutant cDNA was amplified by PCR from pBABE-Stat5a1*6 vector[50] (addgene, 130668) and inserted into the retroviral vector MIT (MSCV-IRES-Thy1.1[51], provided by June-Yong Lee, Yonsei University College of Medicine). High-titer virus was generated by transient transfection of 293FT cell line with retroviral and packaging vectors using TurboFect Transfection Reagent (Thermo Fisher Scientific, R0531). Viral supernatants were harvested at 48 and 72 h after transfection, filtered through a 0.45 μm filter unit (Millipore, SLHVR33RB), and concentrated by centrifugation using Amicon Ultra-15 Centrifugal Filter Unit (Merk, UFC901024).

## BMDC transduction

BMDCs were harvested on day 4, plated ($3 \times 10^6$ cells/well) in 6-well plates, and spin-infected with viral supernatants supplemented with 8 μg/ml polybrene at $1250 \times g$ for 90 min at 32 °C. After the spin, the supernatant was removed and replaced with complete culture medium in the presence of GM-CSF. BMDCs were harvested 48 h post infection and transduction efficiency was examined by analyzing Thy1.1 using flow cytometry.

## RNA sequencing

Strand-specific libraries were generated using the TruSeq PolyA Stranded mRNA sample preparation kit (Ilumina, FC-122-1001) according to the manufacturer's protocol. Barcoded libraries were pooled and sequenced on the Illumina HiSeq2500 generating 100 bp paired-end reads. Three biological replicates were performed for each condition.

## Chromatin immunoprecipitation sequencing (ChIP-seq)

ChIP-seq was performed as previously described[52]. Briefly, chromatin samples prepared using the appropriate number of fixed cells ($3 \times 10^5$ for histone modifications and $5 \times 10^6$ for transcription factors) were sonicated and subsequently immunoprecipitated with each antibody recognizing 5 μg of CTCF (Cell Signaling Technology, 2899), 5 μg of SMC1 (Bethyl lab, A300-055A), 5 μg of RelA (Cell Signaling Technology, 8242), 5 μg of STAT5 (Cell Signaling Technology, 9351), 1 μg of H3K27ac (Abcam, ab4729), 1 μg of H3K4me1 (Abcam, ab8895), 1 μg of H3K4me3 (Abcam, ab8580), or 1 μg of H3K27me3 (Abcam, ab6002). Antibody-chromatin complexes were captured with protein A and G Dynabeads (Invitrogen, 100.02D/100.04D), washed with Low Salt Wash Buffer, High Salt Wash Buffer, and LiCl Wash Buffer. Chromatin-antibody immobilized on magnetic beads were then subjected to tagmentation. Eluted DNA was purified using SPRI Ampure XP beads (Beckman Coulter, A63881) and amplified for 8–12 cycles using Nextera PCR primers. Libraries were purified using dual (0.5x–2.0x) SPRI Ampure XP beads and paired-end sequenced (100 bp) on the Illumina HiSeq2500 platform. Two biological replicates were performed for each condition.

## In situ Hi-C

In situ Hi-C was performed as previously described[3]. In brief, $2 \times 10^6$ BMDCs were crosslinked with 1% formaldehyde (Sigma, F8775) for 10 min and subsequently quenched with 0.125 M glycine (Duchefa Biochemie, G0709). Chromatin was digested using MboI restriction enzyme (New England Biolabs, R0147), followed by biotin incorporation with Biotin-14-dATP (Jena bioscience, NU-835-BIO14-S). After de-crosslinking, ligated DNA was purified and sheared to 200–300 bp. DNA was purified using the MinElute PCR Purification Kit (Qiagen, 28004) and quantified using the Qubit dsDNA HS Assay Kit (Invitrogen, Q32854). Subsequently, 150 ng DNA was used for capture with Dynabeads MyOne Streptavidin C-1 (Invitrogen, 65001), and an appropriate amount of Tn5 enzyme (Illumina, 20034198) was added to the captured DNA to generate the sequencing library. Each library was paired-end sequenced (100 bp) on Illumina Nova-Seq6000 platform. Two biological replicates were performed for each condition.

## HiChIP

HiChIP was performed as previously described[53], using antibodies against 1 μg of H3K27ac (Abcam) or 5 μg of RelA (Cell Signaling Technology). Briefly, cells ($2 \times 10^6$ for H3K27Ac, $1 \times 10^7$ for RelA) were crosslinked with 1% formaldehyde (Sigma) for 10 min and subsequently quenched with 0.125 M glycine (Invitrogen). Chromatin was digested using MboI restriction enzyme (NEB), followed by biotin incorporation with Biotin-14-dATP (Jena bioscience) in end-repair step, ligation, and sonication. Sheared chromatin was then incubated with antibodies recognizing H3K27Ac or RelA at 4 °C overnight. Chromatin-antibody complexes were captured by Protein A and G magnetic bead (Invitrogen) and subsequently washed with Low Salt Wash Buffer, High Salt Wash Buffer, and LiCl Wash Buffers before being eluted. DNA was purified with the MinElute PCR Purification Kit (Qiagen) and quantified using the Qubit dsDNA HS Assay Kit (Invitrogen). Subsequently, 50–150 ng was used for capture with Dynabeads MyOne Streptavidin C-1 (Invitrogen) and an appropriate amount of Tn5 enzyme (Illumina) was added to captured DNA to generate sequencing library. Each library was paired-end sequenced (100 bp) on Illumina NovaSeq6000 platform. Two biological replicates were performed for each condition.

## RNA-seq data processing

Paired end sequencing reads were trimmed using Trim Galore with command-line settings "trim_galore -q 20–trim1–paired" and subsequently aligned to the mouse mm10 genome assembly using STAR[54] with default parameters. The gene expressions were quantified using RSEM[55] and differentially expressed genes were determined using DEseq2[56], with an adjusted p-value threshold of 0.05. The DAVID database[57] was used for pathway and biological process enrichment analysis of differentially expressed genes.

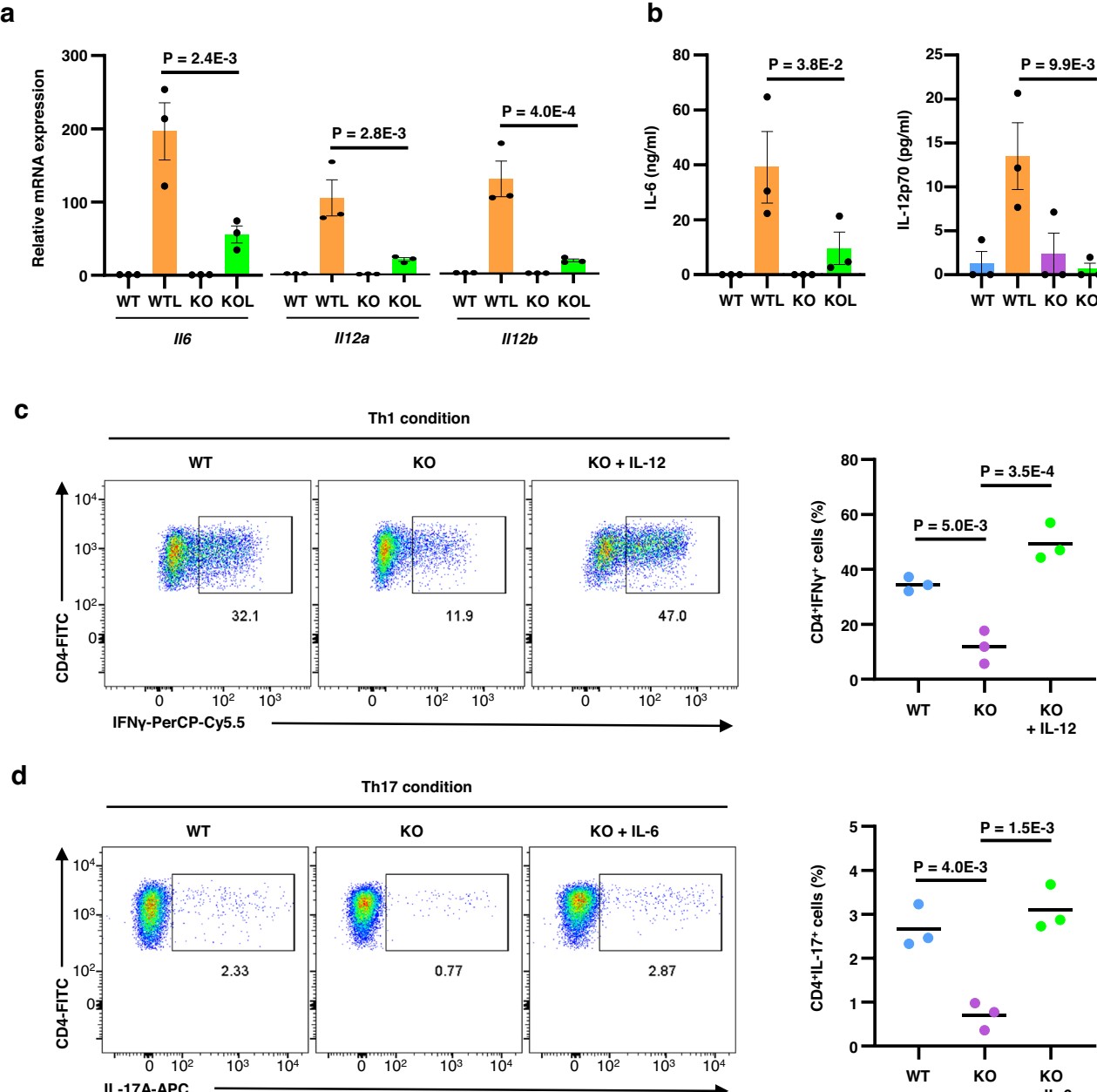

**Fig. 9 | Compromised capacity for inducing Th1 and Th17 CD4⁺ T cells by CTCF-deficient BMDCs. a, b** Decreased expression of IL-6 and IL-12 in mRNA (**a**) and protein (**b**) levels by CTCF depletion when BMDCs were stimulated with LPS for 3 h (**a**) and 24 h (**b**). Error bars show mean ± s.e.m. Significance was calculated using a two-way ANOVA with multiple comparisons of Bonferroni post-test using $n = 3$ independent samples. **c, d** Representative flow cytometric plots (left) and summarized bar graphs (right) showing impaired capacity for inducing Th1 (**c**) and Th17 (**d**) CD4⁺ T cell differentiation by CTCF-deficient BMDCs. In some experiments, IL-12 and IL-6 was additionally added in the Th1 and Th17-driving culture condition, respectively. Error bars show mean ± s.e.m. Significance was calculated using a one-way ANOVA with Tukey's multiple comparisons using $n = 3$ independent samples. Source data are provided as a Source Data file.

## ChIP-seq data processing

Paired-end sequencing reads were trimmed using trim galore with default parameter settings and subsequently aligned to the mouse mm10 genome assembly using bwa (version 0.7.17)[58] with default parameter settings. Duplicate reads were removed using picard tools (version 2.9.2). Peaks were identified for each sample and biological replicate using MACS2 (version 2.2.7.1)[59] with command line options "macs2 callpeak -g mm -f BAMPE –nomodel". Uniquely mapped reads were normalized using deeptools (version 3.3.0)[60] with command line options "–normalizeUsing CPM–binSize 1" to visualize ChIP-seq signals at specific genomic loci by IGV (version 2.8.2)[61]. ChIP-seq heatmaps were generated with deepTools to show normalized read counts at the peak center ±2 kb.

## In situ Hi-C data analysis

Paired-end reads were aligned to mm10 genomes using the HiC-Pro pipeline (version 2.11.1)[62]. Default settings were used to remove duplicate reads, assign reads to MboI restriction fragments, filter for valid interactions, and generate binned interaction matrices. Replicates data were first processed separately. After confirmation of good reproducibility by HiCSpector[63], we merged the replicates and re-processed the data as combined results (Supplementary Table 2). The

data were visualized using Juicebox (version 1.9.8)[64]. The validated contact pairs were transformed to Juicer.hic files with hicpro2juicebox. To segregate A and B compartments, the eigenvector of each chromosome from each sample was generated from the.hic file using Juicer tools (version 1.19.02)[65] 'eigenvector' command, with KR normalization at 100 kb resolution. Topological domain boundaries were identified at a 10 kb resolution based on insulation scores using command-line script "matrix2insulation.pl –b 500000 –ids 200000 –im mean –nt 0.1 –bmoe 3" as previously described[66].

### HiChIP data analysis

Paired-end reads were aligned to mm10 genomes using the HiC-Pro pipeline (version 2.11.1)[62]. Default settings were used to remove duplicate reads, assign reads to MboI restriction fragments, filter for valid interactions, and generate binned interaction matrices. Each replicate data was first processed separately. After confirmation of good reproducibility by HiCSpector, we merged the replicates and re-processed the data as combined results (Supplementary Table 3 and 4). The data were visualized using Juicebox. Loop-calling for the HiChIP experiment was performed using FitHiChIP[67] with 10 kb bin sizes, bias correction by coverage, false discovery rate (FDR) < 0.01 (RelA HiChIP) or $10^{-5}$ (H3K27ac HiChIP), a minimum genomic distance of 20 kb and a maximum genomic distance of 2 Mb.

### Definition of regulatory elements for annotating HiChIP loop anchors

Promoters were defined as ±2.5 kb from the transcription start site (TSS) of each expressed gene. The longest transcript (ENST) from each Ensembl gene id (ENSG) was used to assign unique transcriptional start sites and gene positions. Enhancers were defined as regions with an H3K27ac peak as determined by ChIP-seq. H3K27ac peaks that overlapped a gene promoter were removed from this list. Super-enhancers were defined by applying the ROSE algorithm to H3K27ac peaks with the default stitching size of 12.5 kb[68]. The presence of one or more promoter was considered a promoter HiChIP anchor. The absence of any promoter but presence of enhancer constituted an enhancer HiChIP anchor. The presence of at least one CTCF ChIP-seq but absence of any promoter or enhancer was considered as a CTCF HiChIP anchor.

### Differential analysis of HiChIP loops

For identification of loops with differential strength of chromatin interaction, we considered all H3K27ac HiChIP loops with $q < 10^{-5}$ in at least one of the two conditions being compared. DESeq2 (version 1.24.0)[56] was applied to identify differential loops of H3K27ac HiChIP data using contact counts of each replicate. We selected constant H3K27ac HiChIP loops using a $P > 0.5$ and absolute log2[FC] < 0.378512. Gained and lost H3K27ac HiChIP loops were selected using a $P < 0.1$ and log2[FC] > 0.584963 and log2[FC] < −0.584963, respectively. To identify the loop anchors with differential enhancer activity, we considered all 10 kb-long anchors connecting the significant H3K27ac HiChIP loops with $q < 10^{-5}$ in at least one of the two conditions being compared. DESeq2 (version 1.24.0)[56] was applied to identify differential enhancers using the sum of H3K27ac ChIP-seq peaks normalized by input within each 10 kb-long anchor. For constitutive enhancer, we selected anchors using a $P > 0.5$ and absolute log2[FC] < 0.378512. Activation-inducible and activation-repressed enhancers were selected using a $P < 0.1$ and log2[FC] > 0.378512 and log2[FC] < −0.378512, respectively.

### LOLA enrichment analysis

Anchors of each differential loop were analyzed using LOLA (version 1.8.0)[69] and compared against the LOLA region databases (regionDB) for mm10 to identify enrichment of experimentally derived transcription factor binding locations.

### 3D clique analysis

3D clique analysis was performed following the same procedure as previously reported[70]. In brief, an undirected graph of enhancer-centric chromatin interaction was constructed from H3K27ac HiChIP data where each vertex was a loop anchor and each edge was a significant H3K27ac HiChIP loop. "3D Cliques" were defined by spectral clustering of the H3K27ac-mediated chromatin interaction using cluster_louvain function in igraph R package with default parameters. A 3D clique connectivity was defined as the number of edges connecting vertices within the clique.

### Quantification and statistical analysis

The statistical significance of differences between measurements was determined by Wilcoxon rank sum using the R package, unless otherwise stated. Statistical details of experiments can be found in the figure legends.

### Reporting summary

Further information on research design is available in the Nature Portfolio Reporting Summary linked to this article.

## Data availability

All RNA-seq, ChIP-seq, in situ Hi-C, and HiChIP data generated in this study have been deposited in the GEO under accession number GSE185884. Source data are provided with this paper.

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

## Acknowledgements

This work was supported by the National Research Foundation of Korea (NRF) grants funded by the Korean government (MSIP) (2018M3A9D3079290, 2020R1A2C2013258, and 2022M3A9B6017424 to H.-P.K.) as well as by Yonsei University College of Medicine Faculty Research Grant (6-2017-0079 to H.-P.K.).

## Author contributions

H.K. conceived and supervised the project, oversaw the experimental strategy, computational analysis, and interpretation of data and held overall responsibility for the study. B.Y., E.L., and J.J. conducted the in situ Hi-C, H3K27ac HiChIP. RelA HiChIP, ChIP-Seq, RNA-seq, Immunofluorescence, western experiments, and generated data. W.J., K.K., S.K.#, Y.K., S.O., and K.H.Y. performed bioinformatic data analysis and interpretation. B.Y., S.K.*, T.K., and C.G.P. performed immunological phenotype analysis and interpretation. H.K. and B.Y. wrote the manuscript, with input from all other authors. (S.K.#: Sugyung Kim, S.K.*: Sueun Kim).

## Competing interests

The authors declare no competing interests.
