## [Peer Review File · Nature Communications]

CTCF controls three-dimensional enhancer network underlying the inflammatory response of bone marrow-derived dendritic cellsREVIEWER COMMENTS

Reviewer #1 (Remarks to the Author):

The paper examines GM-DC development in an inducible CTCF depletion model. This system uses the GM-CSF-derived cells used in many studies. The author claim from this (based on the data in Supplemental Figure 2) that CTCF is not required for GM-DC development. However, in the field of dendritic cells, it is now known based on an important Resource Immunity paper published in 2015 that mouse BM cells cultured with GM-CSF alone are heterogeneous, and contain mixtures of monocyte-derived DCs and monocyte-derived macrophages (Helft et al, Immunity 2015). This is not a paper that can be ignored, as it has been cited 279 times in the last 6 years. This study basically transformed the field by making it necessary to evaluate such cultures in terms of what cells might be affected by different conditions.

Thus, in this case, it would be important for the authors to examine whether the CTCF depletion specifically impaired the development of one of these populations, which leads to the overall 50% reduction of total cell number. Since these two populations show completely different phenotypes, for example by FACS analysis, in the expression level of MHC-II and co-stimulatory molecules. In this system, only DCs express co-stimulatory molecules, while MACs do not. Thus, the authors need to examine whether the two populations are individually affected by the loss of CTCF. For example, it could be the case that in the absence of CTCF, one population develops normally, but the other does not develop. Alternately, the impact of CTCF depletion on various genes could be different between the DCs and the MACs. This possible differential effect could also be true for the apoptosis and proliferation assays, and their capacity for antigen uptake and induction of T cell proliferation. These analyses would be necessary to draw the conclusion that "CTCF is dispensable for GM-CSF-mediated differentiation of DC in vitro". Also, throughout the paper, the authors use the term DC. This should be corrected systematically, which in fact is not possible, as GM-DCs are already known to be heterogeneous. Nonetheless, the authors need to replace the "DC" with another more precise term, and I might suggest GM-DC/Macs until they have the new analysis showing the data independently for each type of cell. In summary, the analysis and conclusions need to be reformulated by taking into account the fact that the authors data at present is a bulk analysis of at least two different types of cells.

A second problem here is that of novelty and priority. The claim by the authors that "CTCF is dispensable for higher order chromatin compartment but essential for TAD organization" and that CTCF is required for maintaining "intra-TAD chromatin looping" have already been published in 2017 by Nora et al. in *Cell*, in a slightly different technique using auxin-inducible degron system, and in mouse ES cells. Thus, while the authors do extend the conclusions into another cell type, the problem that the cell type is mixed and may not apply to both DCs and MACs is a problem. The authors here use a tamoxifen-inducible Cre system in GM-DC and get similar conclusions, and while these are not new discoveries, they may be publishable in *Nature Communications*, but should not be allowed to be a sloppy paper by permitting the mixed cell analysis.

Another major problem with the study is that it draws a major conclusion of causality that has not been established. In short, the logical flaw may be that of the classical "true, true, but unrelated" type. Specifically, the authors find two different phenomena, #1) that CTCF depletion results in reduced GM-CSF-induced JAK2/STAT5 signaling and defective LPS-induced NF- κ B activation; and #2) that CTCF depletion results in reduced loop formation and enhancer activation. The authors conclude that phenomenon #1 causes phenomenon #2. However, it is equally possible that the causality is that #2 causes #1, or even that neither causes the other. There is no evidence in the study to draw any conclusion about the cause of either. If the authors wish to draw this conclusion, they need to test it specifically, by, for example, blocking NF κ B signaling by an independent technique, leaving CTCF intact, and then test for the loop formation and enhancer activation. As it is, the authors jump to their big conclusion that "NF- κ B binding facilitates stronger enhancer-promoter interactions as well as higher enhancer activity" and spent two whole pages (lines 269-335) discussing it. Alternately the authors could drop this conclusion and discussion and simply report the independent findings, with a bit of limited speculation and discussion. That approach would not exceed the data.

Finally, in Figure 7c, it is a little misleading for the authors to selectively show gene ontology analysis for only CTCF-dependent RelA target genes. A similar analysis is needed between LPS stimulated WT cells and un-stimulated WT cells, which would allow for the knowledge that whether CTCF deficiency has a selective effect on pro-inflammatory cytokine genes or unbiasedly diminishes all induced gene expression.

Reviewer #2 (Remarks to the Author):

In this well-written manuscript, Yang et al. examined the effects of depletion of the chromatin structural protein CTCF on gene regulation in dendritic cells. They found that CTCF is dispensable for GM-CSF mediated differentiation of dendritic cells, but elegantly showed key structural and transcription effects downstream of CTCF depletion. The authors used these observations of aberrant transcription and structure after CTCF knockout as rationale to further examine chromatin features in dendritic cells. A notable experiment includes measurement of H3K27ac regions plus their relevant chromatin interactions using HiChIP, a very new technique. This experiment revealed chromatin structural links between enhancers and promoters in wild-type cells and those activated with LPS, in addition to CTCF knockout cells. The authors also explored STAT5 binding to understand how JAK/STAT signaling may play a role in transcriptional regulation in the CTCF knockout, and used ChIP-seq to examine RelA (subunit of NF- κ B) binding along the genome and determine its structural context in activated cells (using RelA HiChIP) or in the CTCF knockout (using H3K27ac HiChIP).

This study has important implications within the field of gene regulation, but also within the field of immunology. The methods are very close to the cutting edge of the chromatin field and are clearly written. Overall, the study was modestly descriptive but could be made publishable by addressing experimental concerns below, mainly the question of whether reintroduction of CTCF into CTCF-depleted cells would revert the chromatin structural changes observed in the CTCF knockout.

Major comments:

Could the authors rescue CTCF levels in the knockout cells by reintroducing CTCF via an expression vector? This would answer the question of whether chromatin structure (TADs, insulation boundaries, loops, enhancer-gene interactions) reverts back to normal?

The dispensability of CTCF for activation of BMDCs is an interesting observation. What are the other mechanisms by which BMDC activation is orchestrated? If these mechanisms are inhibited, does CTCF depletion worsen the phenotype? What is the rationale for continuing with the chromatin architecture study if CTCF is dispensable for BMDC activation?

The experimental data from Figure 4 helps answer a key question of whether CTCF mediates distal enhancer loops within TADs in dendritic cells and whether disruption of chromatin structure by CTCF depletion results in cross-TAD interactions and subsequent transcriptional change. Well done.

A supplementary table naming the genes whose interaction and transcription become disrupted with CTCF knockout from Figure 4F would help the reader understand which types/classes of genes have their structure and transcription preserved by CTCF in the normal context. Is *Aldh1a2* an important gene within the context of the authors' studies?

Does the normal expression of the two genes (*Trim25* and *Irak2*) in Figure 5H get rescued if the STAT5 is overexpressed in CTCF knockout cells? This would help determine whether STAT5 activity can override CTCF depletion-mediated chromatin structural defects to preserve normal transcription of CTCF-dependent genes.

The bottom of Supplementary Figure 6 and Figure 7F provide specific evidence to suggest that RelA acts not only as a transcription factor, but as a factor found at enhancers that can may affect transcription. Does this RelA HiChIP interaction profile not appear within wild type cells (not LPS treated)? What was the rationale for only measuring chromatin structure via HiChIP in the LPS treated wild-type cells only?

Figure 8 shows an interesting observation. Since the manuscript focuses on chromatin structure and gene regulation in dendritic cells, the authors may consider examining chromatin structure at IL12 and IL6 downstream target gene loci in CTCF knockout cells and in CTCF knockout + IL12/6 cells, to understand what is going on at these loci when additional IL12 or IL6 is introduced into the system.

Minor comments:

To ensure the reproducibility of the results, the authors should include catalog numbers for all relevant antibodies/reagents used in this study.

Signal quantifications and molecular weight markers, in addition to an indication of the n for each Western blot, would be useful.

In general, graph titles would be helpful for those that want to quickly examine the figures.

How were the ChIP-seq tracks generated?

Representative region shown in Figure 1E is a good starting example of efficient CTCF depletion, although the authors have space to show additional examples at relevant gene loci here.

The effects of CTCF depletion on TADs/boundaries/loops have been examined in Zuin et al. (PMID: 24335803) and Rosa-Garrido et al. (PMID: 28802249), and these two studies seem to be good ones to reference within the context of the current manuscript.

In Supplementary Figure 3A, the authors could probably show the same maximum values within the different groups of 4 heatmaps, to show more clearly that TAD structure is weakened with the CTCF knockout.

For a couple of columns in Supplemental Figure 4A-B, the numbers add up to greater than 100%. Please clarify these figures.

Some rationale for gene selection for Figure 3F would help the reader understand why each gene's interaction/transcription change is important for the resultant phenotype after LPS.

How do the authors define loops gained in CTCF knockout in Figure 4? Are they still examining HiChIP data?

Figure 4H should have an indication of the n for the experiment, and Figure 4I needs molecular weight markers and an indication of how many blots were performed.

Supplemental Figure 5E does not show a log₂(fold-change). Is this an RT-qPCR experiment?

The analysis in Figure 5G is very useful to the reader and is a great example of how to dive deeper into Gene Ontology analyses. Are these all the genes in the "Regulation of NF-κB activity" term? If not, it would be useful to see all of them together on the graph.

Figure 6 could use extra labeling to indicate that Figure 6 I-J shows H3K27ac loops. A brief description of the LOLA algorithm would be useful here as well.

Why are the FDR significance thresholds different for the significant loops between the H3K27ac and

RelA HiChIP experiments? Some commentary on this would provide deeper insight into the technical details of these experiments.

Figures 7G-H are not called out in the main text.

Point-by-point responses to the reviewers' comments

COMMENTS FROM REVIEWER #1

Comment #1:

The paper examines GM-DC development in an inducible CTCF depletion model. This system uses the GM-CSF-derived cells used in many studies. The author claim from this (based on the data in Supplemental Figure 2) that CTCF is not required for GM-DC development. However, in the field of dendritic cells, it is now known based on an important Resource Immunity paper published in 2015 that mouse BM cells cultured with GM-CSF alone are heterogeneous, and contain mixtures of monocyte-derived DCs and monocyte-derived macrophages (Helft et al, Immunity 2015). This is not a paper that can be ignored, as it has been cited 279 times in the last 6 years. This study basically transformed the field by making it necessary to evaluate such cultures in terms of what cells might be affected by different conditions. Thus, in this case, it would be important for the authors to examine whether the CTCF depletion specifically impaired the development of one of these populations, which leads to the overall 50% reduction of total cell number. Since these two populations show completely different phenotypes, for example by FACS analysis, in the expression level of MHC-II and co-stimulatory molecules. In this system, only DCs express co-stimulatory molecules, while MACs do not. Thus, the authors need to examine whether the two populations are individually affected by the loss of CTCF. For example, it could be the case that in the absence of CTCF, one population develops normally, but the other does not develop. Alternately, the impact of CTCF depletion on various genes could be different between the DCs and the MACs. This possible differential effect could also be true for the apoptosis and proliferation assays, and their capacity for antigen uptake and induction of T cell proliferation. These analyses would be necessary to draw the conclusion that “CTCF is dispensable for GM-CSF-mediated differentiation of DC in vitro”. Also, throughout the paper, the authors use the term DC. This should be corrected systematically, which in fact is not possible, as GM-DCs are already known to be heterogeneous. Nonetheless, the authors need to replace the “DC” with another more precise term, and I might suggest GM-DC/Macs until they have the new analysis showing the data independently for each type

of cell. In summary, the analysis and conclusions need to be reformulated by taking into account the fact that the authors data at present is a bulk analysis of at least two different types of cells.

Response #1:

We thank the reviewer for addressing this important point. As suggested, we performed FACS analysis to evaluate the cellular heterogeneity of the “*in vitro* BMDC differentiation model” used in the study. Consistent with the previous report by Helft *et al.*¹, the CD11c⁺MHCII⁺ cells from our GM-CSF BM cultures comprised two sub-populations, but with different cellular proportions: 30% MHCII^{int}CD11b^{high} (which we believe that corresponded to the GM-MACs population identified by Helft *et al.*¹) and approximately 60% MHCII^{high}CD11b^{int}. Noteworthy, most MHCII^{high}CD11b^{int} cells were CD115⁻CD135⁺ cells (which we believe that corresponded to the GM-DC population identified by Helft *et al.*¹), whereas very few of these cells were CD115⁻CD135⁻ (which we believe that corresponded to the GM-DN cells identified by Helft *et al.*¹).

In addition to the argument raised by Helft *et al.* and others^{1,2}, there are several other opinions about the usefulness of murine BM-DC cultures in the study of DC biology^{3,4}. For example, Menges *et al.* demonstrated that MHC II^{low} cell populations contain progenitors for both macrophages (which downregulate MHC II from their surface) and immature DCs (which further upregulate MHC II on their surface, an indication of maturation)³. Moreover, it was reported that IL-4 may drive the maturation of dendritic cells⁵⁻⁸. In this regard, we investigated whether IL-4 supplementation could impact on the cellular heterogeneity of GM-CSF BM cell cultures. Interestingly, the fraction of MHCII^{int}CD11b^{high} cells (GM-MACs) among the CD11c⁺MHCII⁺ population arising in “GM-CSF + IL-4 culture” were substantially downregulated to around 10%, whereas more than 80% of CD11c⁺MHCII⁺ cells were present as MHCII^{high}CD11b^{int}, with most of them being CD115⁻CD135⁺ cells (GM-DCs). These results strongly suggest that the MHCII^{int}CD11b^{high} (GM-MACs) population in our GM-CSF BM cell cultures actually comprised a significant fraction of immature DCs that could potentially differentiate into GM-DCs. It should also be noted that the proportion of CD11c⁺MHCII⁺CD11b^{int} cells was almost equivalent between the WT and KO GM-CSF BM cultures regardless of IL-4

supplementation (Supplementary Fig. 1), indicating that CTCF depletion does not significantly affect the cellular heterogeneity of GM-CSF BM culture.

It remains unclear why the extents of BM differentiation and the resulting cellular heterogeneity observed in the GM-CSF BM cultures varied among research groups. However, considering a report demonstrating that the extent of BM differentiation may change in a GM-CSF dose-dependent manner⁹, we speculate that the GM-CSF used in our study might provide more favorable differentiation conditions toward DC populations with minor fraction of macrophage cells.

Taken all these findings together, we believe that our “*in vitro* differentiated BMDCs,” comprising mainly DCs, including GM-DCs and potentially immature DCs, is suitable for the study of DC biology. Therefore, we cordially suggest to retain the term “dendritic cells” in the manuscript as is. However, considering the controversy over cellular identity and heterogeneity of GM-CSF BM cultures^{4,10}, we would rather delete the statement describing that “CTCF is dispensable for GM-CSF-mediated differentiation of dendritic cells *in vitro*.” Instead, we would simply demonstrate that the cellular proportion of our GM-CSF BM cultures was not affected by CTCF depletion (Supplementary Fig. 1). We have now added these new data to the revised version of the manuscript and hope that this explanation clarifies the reviewer’s point.

Supplementary Figure 1. (related to Figure 1) Phenotype of cells developing in GM-CSF BM cultures. **a** Phenotype of representative GM-CSF BM cultures with or without IL-4 at day 6. CD11c⁺MHCII⁺ cells can be sub-divided on the basis of CD11b and MHCII expression (left), which can further be sub-divided on the basis of CD135 and CD115 expression (right). Boxes depict gates and numbers correspond to percentage of cells in each gate. Data are representative of three independent experiments **b** Summarized bar graph for percentage of CD11c⁺MHCII^{high}CD11b^{int} BMDCs. Error bars represent mean \pm standard error of the mean (s.e.m).

Comment #2:

A second problem here is that of novelty and priority. The claim by the authors that “CTCF is dispensable for higher order chromatin compartment but essential for TAD organization” and that CTCF is required for maintaining “intra-TAD chromatin looping” have already been published in 2017 by Nora et al. in Cell, in a slightly different technique using auxin-inducible degron system, and in mouse ES cells. Thus, while the authors do extend the conclusions into another cell type, the problem that the cell type is mixed and may not apply to both DCs and MACs is a problem. The authors here use a tamoxifen-inducible Cre system in GM-DC and get similar conclusions, and while these are not new discoveries, they may be publishable in Nature Communications, but should not be allowed to be a sloppy paper by permitting the mixed cell analysis.

Response #2:

We thank the reviewer for this comment. Indeed, the claim that “CTCF is dispensable for higher order chromatin compartment but essential for TAD organization” and that “CTCF is required for maintaining intra-TAD chromatin looping” is not novel; accordingly, we have already cited several relevant papers in the Introduction and Discussion sections of the manuscript. Nonetheless, our study aimed to dissect the mechanism through which CTCF regulates the complex molecular events occurring in mouse primary DCs in response to pathogenic stimuli by analyzing the global transcriptomic, epigenetic, and topologic changes. Indeed, consistent with previous reports that used an auxin-inducible degron system to deplete CTCF in several cellular models ¹¹⁻¹³, we also demonstrated that cre/lox-mediated depletion of CTCF in mouse primary DCs retains the compartment organization but compromises the global TAD insulation. We again cited the relevant related references in the corresponding part of Results. Noteworthy, however, the key contribution of our study is the mapping of 3D enhancer networks underlying the activation of DCs. The limited Hi-C resolution usually prevents the robust identification of enhancer-promoter interactions. To overcome this issue, we performed H3K27ac HiChIP to generate high-resolution genome-wide contact maps of active enhancers and target genes in BMDCs. Moreover, we investigated the impact of NF- κ B activation on the 3D genome organization and we have identified the

role of RelA-mediated long-range chromatin interactions in the optimal immune function of DCs. With these comprehensive approaches, we provide novel insights into how three-dimensional enhancer networks control gene expression during DC activation and offer an integrative view of the complex activities of CTCF in the inflammatory response of DCs. We hope that this explanation clarifies the reviewer's point.

Comment #3:

Another major problem with the study is that it draws a major conclusion of causality that has not been established. In short, the logical flaw may be that of the classical “true, true, but unrelated” type. Specifically, the authors find two different phenomenon, #1) that CTCF depletion results in reduced GM-CSF-induced JAK2/STAT5 signaling and defective LPS-induced NF-kB activation; and #2) that CTCF depletion results in reduced loop formation and enhancer activation. The authors conclude that phenomenon #1 causes phenomenon #2. However, it is equally possible that the causality is that #2 causes #1, or even that neither causes the other. There is no evidence in the study to draw any conclusion about the cause of either. If the authors wish to draw this conclusions, they need to test it specifically, by, for example, blocking NF-kB signaling by an independent technique, leaving CTCF intact, and then test for the loop formation and enhancer activation. As it is, the authors jump to their big conclusion that “NF-kB binding facilitate stronger enhancer-promoter interactions as well as higher enhancer activity” and spent two whole pages (lines 269-335) discussing it. Alternately the authors could drop this conclusion and discussion and simply report the independent findings, with a bit of limited speculation and discussion. That approach would not exceed the data.

Response #3:

We thank the reviewer for these critical comments. As requested, we blocked NF-kB signaling using JSH-23, which prevents the nuclear translocation of NF-kB RelA¹⁴ while leaving CTCF intact (Fig. 8a and 8b), and then explored whether activation of NF-kB signaling could facilitate enhancer activation and loop formation (Fig. 8c–8f). First, H3K27ac ChIP-seq analysis demonstrated that enrichment of H3K27ac at RelA binding sites induced by LPS stimulation was almost completely abrogated by JSH-23 pre-

treatment (Fig. 8c). Next, the RelA-mediated chromatin interactions (shown in Fig. 7a) were subject to differential H3K27ac HiChIP loop analysis, which revealed 346 gained loops and 295 lost loops due to LPS stimulation (Supplementary Fig. 10a). Interestingly, these dynamic changes in RelA-mediated chromatin interactions induced by LPS stimulation were significantly attenuated by pre-treatment of the BMDCs with JSH-23 (Fig. 8d). Further, RelA target genes (shown in Supplementary Fig. 9c) were analyzed by RNA-seq, which revealed 827 upregulated and 406 downregulated genes in response to LPS stimulation (Supplementary Fig. 10b). The extent of changes in the transcript abundance of these differentially expressed RelA target genes was significantly diminished by pre-treatment of the BMDCs with JSH-23 before LPS stimulation (Fig. 8e). The best examples showing the effect of NF- κ B inhibition on enhancer activation, loop formation, and the resultant RNA expression profile were provided by Il6 and Il12a (Fig. 8f). LPS stimulation resulted in increased mRNA expression accompanied by enhanced H3K27ac levels upstream of Il6 and Il12a promoters, which were severely abrogated by pre-treatment of the BMDCs with JSH-23 (Fig. 8f). Moreover, V4C plot for H3K27ac HiChIP loop with the Il6 and Il12a promoter regions as anchor demonstrated that loop formation induced by LPS stimulation between promoters and RelA binding sites were also decreased by JSH-23 pre-treatment (Fig. 8f). Taken together, these results indicated that NF- κ B signaling can control spatial enhancer-promoter proximity, as well as distal enhancer activities for the optimal expression of its target genes. We added these new data to the revised version of the manuscript (Fig. 8 and Supplementary Fig. 10), and hope that these additional results clarify the reviewer's point.

Figure 8. NF- κ B signaling controls enhancer activation and enhancer-promoter interactions for optimal expression of its target genes. **a** WT BMDCs were either untreated or treated with JSH-23 for 3 h and Western blot analysis was performed to assess LPS-induced RelA nuclear translocation. The data are representative of three independent experiments with similar results. **b** Densitometric analysis of the proteins in (a) was performed using ImageJ. **c** Heatmaps of normalized ChIP-seq tag densities for H3K27ac at RelA binding sites called from LPS-stimulated WT BMDCs (WTL). Histogram showing the average tag density of ChIP-seq peaks are displayed at the right of the heatmaps. The data are representative of two independent experiments. **d** The effect of JSH-23 on the extent of changes in the chromatin interactions of either gained or lost RelA loops in response to LPS stimulation was analyzed by H3K27ac HiChIP. **e** The effect of JSH23 on the transcript abundance of either upregulated or downregulated RelA target genes in response to LPS stimulation was analyzed by RNA-seq. DEG: differentially expressed gene. **f** Snapshot showing virtual 4C plots (V4C), ChIP-seq signal tracks, and RNA-seq signal tracks (from top to bottom) at the *Il6* (**left**) and *Il12a* (**right**) loci. Sky blue, blue, orange, and brown represent unstimulated WT BMDCs in the absence of JSH-23 (WT) or pretreated with JSH-23 (WT+JSH23), LPS-stimulated WT BMDCs in the absence of JSH-23 (WTL) or pretreated with JSH-23 (WTL+JSH23), respectively. V4C shows normalized H3K27ac HiChIP contact counts with the TSSs of *Il6* and *Il12a* as the viewpoint. Gray vertical bars highlight the location of the viewpoints.

Supplementary Figure 10. (related to Figure 8) NF- κ B signaling control enhancer activation and enhancer-promoter interaction for the optimal expression of its target genes. a Volcano plot of H3K27ac HiChIP of untreated WT BMDCs versus LPS-stimulated WT BMDCs (WTL) showing significant changes in RelA-mediated chromatin interactions in response to LPS stimulation. The number of loops exhibiting >1.5 -fold increases in WT (blue) or WTL (red) BMDCs with a P-value <0.1 is indicated. **b** RNA-seq MA plot of untreated WT BMDCs versus LPS-stimulated WT BMDCs (WTL) for showing significant changes in the transcript abundance of RelA target genes in response to LPS stimulation. The number of genes exhibiting >2 -fold increases in WT (blue) or WTL (red) BMDCs with a false discovery rate <0.05 is indicated.

Comment #4:

Finally, in Figure 7c. it is a little misleading for the authors to selectively show gene ontology analysis for only CTCF-dependent RelA target genes. A similar analysis is needed between LPS stimulated WT cells and un-stimulated WT cells, which would allow for the knowledge that whether CTCF deficiency has a selective effect on pro-inflammatory cytokine genes or unbiased diminishes all induced gene expression.

Response #4:

As requested, we analyzed RNA-seq data again to explore whether CTCF depletion had any selective effect on the LPS-induced gene expression. Out of 1,579 genes whose expression was upregulated by LPS stimulation in WT BMDCs (WTL vs. WT) (Fig. 3d), 403 were significantly downregulated by CTCF depletion after LPS stimulation (KOL vs. WTL) (Supplementary Fig. 9e). Gene ontology analysis showed that these 403 genes were enriched in positive regulation of cytokine production and inflammatory-related pathways (Supplementary Fig. 9f). These data demonstrated that CTCF depletion in BMDCs compromises the expression of genes significantly related with inflammation-associated pathways, including pro-inflammatory cytokine production. We added these new data to the revised version of the manuscript (Supplemental Fig. 9e and 9f).

Supplementary Figure 9. (related to Figure 7) Mapping NF- κ B-bound chromatin loops in LPS-stimulated WT BMDCs. e Scatter plot showing log₂-fold changes in RNA expression levels of total genes. The number of genes exhibiting >2-fold changes in WTL versus WT and 1.5-fold changes in KOL versus WTL with a false discovery rate <0.05 is indicated at each quartile. **f** Bar plot of $-\log_{10}$ P value showing enrichment of gene ontology terms on the genes whose expressions were upregulated by LPS stimulation and downregulated by CTCF depletion.

COMMENTS FROM REVIEWER #2

In this well-written manuscript, Yang et al. examined the effects of depletion of the chromatin structural protein CTCF on gene regulation in dendritic cells. They found that CTCF is dispensable for GM-CSF mediated differentiation of dendritic cells, but elegantly showed key structural and transcription effects downstream of CTCF depletion. The authors used these observations of aberrant transcription and structure after CTCF knockout as rationale to further examine chromatin features in dendritic cells. A notable experiment includes measurement of H3K27ac regions plus their relevant chromatin interactions using HiChIP, a very new technique. This experiment revealed chromatin structural links between enhancers and promoters in wild-type cells and those activated with LPS, in addition to CTCF knockout cells. The authors also explored STAT5 binding to understand how JAK/STAT signaling may play a role in transcriptional regulation in the CTCF knockout, and used ChIP-seq to examine RelA (subunit of NF- κ B) binding along the genome and determine its structural context in activated cells (using RelA HiChIP) or in the CTCF knockout (using H3K27ac HiChIP).

This study has important implications within the field of gene regulation, but also within the field of immunology. The methods are very close to the cutting edge of the chromatin field and are clearly written. Overall, the study was modestly descriptive but could be made publishable by addressing experimental concerns below, mainly the question of whether reintroduction of CTCF into CTCF-depleted cells would revert the chromatin structural changes observed in the CTCF knockout.

→ **We sincerely thank the reviewer for the careful review of our manuscript.**

Comment #1:

Could the authors rescue CTCF levels in the knockout cells by reintroducing CTCF via an expression vector? This would answer the question of whether chromatin structure (TADs, insulation boundaries, loops, enhancer-gene interactions) reverts back to normal?

Response #1:

We thank the reviewer for highlighting this important point, and we completely agree that the rescue of CTCF in CTCF knockout cells to validate its crucial role in hierarchical 3D chromatin organization is important. In this study, depletion of CTCF in mouse primary dendritic cells was accomplished using tamoxifen-inducible *Ctcf* conditional knockout BM cells (CreER;*CTCF^{fl/fl}*). As suggested, we tried to reintroduce CTCF into CTCF-deficient cells via retrovirus infection, by which we were able to overexpress *Stat5* in CTCF knockout cells (please see the response #5 to reviewer #2). To generate the retroviruses, we transfected retroviral vectors into 293FT packaging cells as previously described¹⁵. The control empty vector (MSCV-IRES-Thy1.1)¹⁶ and *Stat5*-expression vector (MSCV-*Stat5a1*6*-IRES-Thy1.1) exhibited very high transfection efficiency that led to high virus titers (lanes 1 and 2 in the Figure below), confirming the validity of our protocol for production of retroviruses. However, the CTCF-expressing vector (MSCV-3XFlag-CTCF-IRES-Thy1.1) showed very low transfection efficiency and failed to produce enough number of retroviruses (lane 3 in the Figure below). Another CTCF-expression vector (MSCV-3XFlag-CTCF-eGFP-IRES-Thy1.1) generated by cloning the 3XFlag-CTCF-eGFP fragment from the pKS004-pCAGGS-3XFlag-CTCF-eGFP¹⁷ vector into the MSCV-IRES-Thy1.1 control vector again showed low transfection efficiency (lane 4 in the Figure below). Moreover, other CTCF expression vectors (pMY-CTCFbiotag-T2A-mOrange¹⁸ and pKS004-pCAGGS-3XFlag-CTCF-eGFP¹⁷), which were previously reported and available from Addgene, also failed to show high transfection efficiency (lanes 5 and 6 in the Figure below). We have no definite explanation as to why all the CTCF-expression vectors showed very low transfection efficiencies to generate retrovirus; nonetheless, we may speculate that overexpression of CTCF, unlike *Stat5*, was somehow toxic to the packaging cells.

As an alternative way for depleting and rescuing CTCF, we could consider an auxin-

inducible degron system that allows rapid and specific depletion of a mAID-tagged protein via proteasome-dependent degradation upon auxin treatment¹¹⁻¹³. For example, Nora *et al.*¹³ generated the auxin-inducible CTCF degron system in mouse embryonic stem cells and showed that acute depletion of endogenous CTCF in ESCs disrupted looping between CTCF target sites and insulation of TADs. They also demonstrated that restoring CTCF, by washing off auxin, reinstated the proper architecture on altered chromosomes, indicating a powerful instructive function for CTCF in chromatin folding. We regret that we could not rescue CTCF in CTCF-deficient dendritic cells due to low transfection efficiency, but we hope that the findings reported by Nora *et al.* may help answer the reviewers' question regarding whether reintroduction of CTCF into CTCF-depleted cells could revert the chromatin structural changes observed in the CTCF knockout cells. Generation of auxin-inducible CTCF degron system to explore the role of CTCF in mouse primary dendritic cells would be a quite attractive project, but it is not feasible right now and would be beyond the scope of our current manuscript.

	Vector	Purpose	Reference	Addgene catalog #
1	MSCV-IRES-Thy1.1	Retroviral vector	Wu et al., 2006, Cell	#17442
2	MSCV-Stat5a1*6-IRES-Thy1.1	Retroviral vector	This study	-
3	MSCV-3XFlag-CTCF-IRES-Thy1.1	Retroviral vector	This study	-
4	MSCV-3xFlag-CTCF-eGFP-IRES-Thy1.1	Retroviral vector	This study	-
5	pMy-CTCFbiotag-T2A-mOrange	Retroviral vector	Nakahashi et al., 2013, Cell Reports	#50564
6	pKS004-pCAGGS-3XFlag-CTCF-eGFP	Expression vector	Nora et al., 2020, Nature Communications	#156438

[Representative flow cytometric plots of the frequency of transfection efficiency using the indicated vectors.]

Comment #2:

The dispensability of CTCF for activation of BMDCs is an interesting observation. What are the other mechanisms by which BMDC activation is orchestrated? If these mechanisms are inhibited, does CTCF depletion worsen the phenotype? What is the rationale for continuing with the chromatin architecture study if CTCF is dispensable for BMDC activation?

Response #2:

We thank the reviewer for this comment. In this study, we used an “*in vitro* BMDC differentiation model” to investigate the function of CTCF in the three-dimensional enhancer network underlying the inflammatory response of mouse primary DCs. Since DCs undergo several maturation events upon pathogenic stimuli, we investigated whether our GM-CSF mouse BM cultures were suitable to study the DC biology by analyzing several typical characteristics of DCs, such as antigen uptake, induction of T cell proliferation, and activation of costimulatory molecules (Supplementary Fig. 3). Given that CTCF depletion did not change these general features of DCs, we believe that *in vitro*-differentiated BMDCs are a useful model to investigate the mechanism through which CTCF regulates the complex molecular events during DC activation. Moreover, as mentioned by reviewer #1 (comment #1), there is controversy on the “DC-ness” of murine BM-DCs^{1,2,4,10}. In response to this comment, we performed flow cytometry analysis and demonstrated that the GM-CSF mouse BM cultures generated in our study comprise mainly GM-DCs (defined as CD11c⁺MHCII^{high}CD11b^{int} CD115⁻CD135⁺) and some immature DCs (contained in the MHCII^{int}CD11b^{high}) populations regardless of CTCF depletion (Supplementary Fig. 1). These results again validate the usefulness of our “*in vitro* BMDC differentiation model” for addressing DC biology. However, we would rather delete the statement describing that “CTCF is dispensable for GM-CSF-mediated differentiation of dendritic cell *in vitro*,” considering the controversy over cellular identity and heterogeneity of GM-CSF BM cultures^{4,10}. We hope that this explanation clarifies this reviewer’s point.

Comment #3:

The experimental data from Figure 4 helps answer a key question of whether CTCF mediates distal enhancer loops within TADs in dendritic cells and whether disruption of chromatin structure by CTCF depletion results in cross-TAD interactions and subsequent transcriptional change. Well done.

Response #3:

We thank the reviewer for the appreciation of our work and the positive feedback.

Comment #4:

A supplementary table naming the genes whose interaction and transcription become disrupted with CTCF knockout from Figure 4F would help the reader understand which types/classes of genes have their structure and transcription preserved by CTCF in the normal context. Is *Aldh1a2* an important gene within the context of the authors' studies?

Response #4:

We thank the reviewer for the suggestions. By combining *in situ* Hi-C, H3K27ac HiChIP, and RNA-seq data generated from WT and CTCF-deficient BMDCs, we were able to identify 123 genes which expression was upregulated, possibly due to the augmented enhancer-promoter interactions established *de novo* within the same TADs in the CTCF-deficient BMDCs (Fig. 4d–4f). As requested, the list of these genes was presented as Supplementary Table 5.

***Aldh1a2*, the third most upregulated gene among the identified genes, encodes the aldehyde dehydrogenase 1A2 enzyme that catalyzes the synthesis of retinoic acid from retinaldehyde¹⁹. Given the critical role of retinoic acid in the induction and suppression of Treg and Th17 differentiation²⁰⁻²², respectively, increased expression and enhanced enzyme activity of *Aldh1a2* in CTCF-deficient BMDCs (Fig. 4h–4j) may provide favorable conditions for the development of T cell tolerance, which will be explored in our**

next project to further support the distinct pathophysiological role of CTCF in DC-mediated immune responses.

Comment #5:

Does the normal expression of the two genes (*Trim25* and *Irak2*) in Figure 5H get rescued if the STAT5 is overexpressed in CTCF knockout cells? This would help determine whether STAT5 activity can override CTCF depletion-mediated chromatin structural defects to preserve normal transcription of CTCF-dependent genes.

Response #5:

We thank this reviewer for addressing this important point. One of the most interesting findings of our study was that depletion of CTCF attenuates the GM-CSF-mediated JAK2/STAT5 signals (Fig. 5a and 5b). Moreover, we identified 476 genes, including *Trim25* and *Irak2*, whose expression was influenced by defective JAK2/STAT5 signaling (Fig. 5c–5e). As requested, we introduced a constitutively-active STAT5 variant (STAT5a1*6)^{23,24} into CTCF-deficient BMDCs by retrovirus infection (Supplementary Fig. 7a) to explore whether overexpression of STAT5 activity could override the CTCF depletion-mediated chromatin structural defects. Western blot analysis demonstrated that phosphorylation of STAT5 was much higher in KO BMDCs infected with STAT5a1*6 virus compared with those infected with the control virus, indicating successful rescue of Stat5 activity in CTCF-deficient cells (Supplementary Fig. 7b). However, the rescue of Stat5 activity in CTCF-deficient BMDCs did not increase the mRNA expression of *Trim25* and *Irak2*. These results suggested that downregulation of these Stat5 target genes in CTCF-deficient BMDCs was not simply due to attenuated Stat5 signaling, and that restoration of STAT5 activity was not sufficient to overcome the disrupted looping between enhancers and their target genes observed in CTCF-deficient BMDCs (Supplementary Fig. 7c). We added these new data to the revised version of the manuscript (Supplementary Fig. 7).

Supplementary Figure 7. (related to Figure 5) Restoration of STAT5 activity in CTCF-deficient BMDCs. a Schematic showing MSCV retroviral vector expressing constitutively-active STAT5 variant (STAT5a1*6) or empty vector (MIT). **b** KO BMDCs were infected with retrovirus and Western blotting was performed with indicated antibodies. The data were representative of three independent experiments with similar results. **c** Relative mRNA expression levels for *Trim25* and *Irak2*. Three biological replicates were performed for each condition. Error bars represent mean \pm standard error of the mean (s.e.m). Significance in (c) was calculated using an unpaired Mann-Whitney two-tailed *t*-test. ns, not significant.

Comment #6:

The bottom of Supplementary Figure 6 and Figure 7F provide specific evidence to suggest that RelA acts not only as a transcription factor, but as a factor found at enhancers that can may affect transcription. Does this RelA HiChIP interaction profile not appear within wild type cells (not LPS treated)? What was the rationale for only measuring chromatin structure via HiChIP in the LPS treated wild-type cells only?

Response #6:

We thank the reviewer for addressing this point. HiChIP combines Hi-C and ChIP-seq and provides an efficient method to identify chromatin looping events mediated by specific transcription factors. Translocation of NF- κ B from the cytoplasm to the nucleus is a critical step in the coupling of extracellular stimuli to the transcriptional activation of specific target genes. Immunofluorescence and Western blot analysis demonstrated that the nuclear translocation of RelA following LPS stimulation was considerably reduced by CTCF depletion (Fig. 6b–6d). Moreover, ChIP-seq analysis revealed that LPS stimulation potently induced genome-wide chromatin binding of RelA in WT BMDCs but not in KO BMDCs (Fig. 6e–6g). Furthermore, the number of H3K27ac HiChIP loops overlapping with RelA ChIP-seq peaks at least at one of the loop anchors dramatically increased upon LPS stimulation in WT BMDCs but not in KO BMDCs (Fig. 6l). Given that the concentration of chromatin-bound transcription factor is the key factor for successful HiChIP analysis, we performed RelA HiChIP analysis only with LPS-stimulated WT BMDCs, where enough amount of nuclear RelA was present so that efficient RelA enrichment was possible. However, we believe other conditions, including untreated WT BMDCs, untreated KO BMDCs, and LPS-stimulated KO BMDCs, were not suitable for HiChIP analysis because of the very low amount of nuclear RelA in these cells. We hope that this explanation clarifies the reviewer’s point.

Comment #7:

Figure 8 shows an interesting observation. Since the manuscript focuses on chromatin structure and gene regulation in dendritic cells, the authors may consider examining chromatin structure at IL12 and IL6 downstream target gene loci in CTCF knockout cells and in CTCF knockout + IL12/6 cells, to understand what is going on at these loci when additional IL12 or IL6 is introduced into the system.

Response #7:

We thank the reviewer for raising this interesting point. Naïve CD4⁺ T cells differentiate into diverse effector and regulatory subsets to orchestrate immunity and tolerance^{25,26}. In addition to T cell intrinsic signals, the innate immune system actively instructs adaptive

immunity through antigen presentation and immunoregulatory cytokine production^{25,26}. For example, DC-producing IL-12 supports Th1 development²⁷, whereas DC-producing IL-6 supports Th17 development²⁸. Given that production of IL-6 and IL-12 was impaired in CTCF-deficient BMDCs, we examined the capacity of CTCF-deficient BMDCs for T cell differentiation in *in vitro* BMDC/CD4 T cell-coculture conditions. Indeed, the reduced T cell differentiation was largely attributed to defective cytokine production in the CTCF-deficient BMDCs, since treatment with exogenous IL-12 and IL-6 significantly rescued Th1 and Th17 differentiation in CTCF-deficient BMDCs, respectively (Fig. 9c and 9d). As pointed out by this reviewer, the chromatin structure at *Il12* and *Il6* downstream target gene loci, such as *Ifng* and *Il17* loci, during CD4⁺ T cell differentiation has long been explored by many groups²⁹⁻³¹. We agree that studying the effect of CTCF depletion on chromatin structure in CD4⁺ T cells is quite interesting and important. However, we sincerely hope that this reviewer appreciates the wealth of data and analyses provided and concurs that answering this question using CTCF-deficient CD4⁺ T cell is beyond the scope of our current manuscript.

minor review

Comment #8:

To ensure the reproducibility of the results, the authors should include catalog numbers for all relevant antibodies/reagents used in this study.

Response #8:

We would like to thank the reviewer for this suggestion. We have now described the catalogue numbers for all relevant antibodies and reagents in the Methods section.

Comment #9:

Signal quantifications and molecular weight markers, in addition to an indication of the n for

each Western blot, would be useful.

Response #9:

We thank the reviewer for the suggestion. Now, the signal quantifications, molecular weight markers, and the number of biological replicates for each Western blot have been described (Fig. 1c, 4i, 5a, 6a, 6d, and 8a)

Comment #10:

In general, graph titles would be helpful for those that want to quickly examine the figures.

Response #10:

We thank the reviewer for this suggestion. We have now labeled the graph titles for some of the figures, including Fig. 3a, 3b, 3c, 4a, 8d, and 8e.

Comment #11:

How were the ChIP-seq tracks generated?

Response #11:

Uniquely mapped reads were normalized using deepTools (version 3.3.0) with command line options “--normalizeUsing CPM --binSize 1” to visualize ChIP-seq signal at specific genomic loci by IGV (version 2.8.2). ChIP-seq heatmaps were generated using deepTools to show normalized read counts at the peak center ± 2 kb. This information has been described in the “ChIP-Seq Data Processing” subsection of the Methods section of the revised manuscript.

Comment #12:

Representative region shown in Figure 1E is a good starting example of efficient CTCF depletion, although the authors have space to show additional examples at relevant gene loci here.

Response #12:

As suggested, additional examples of efficient depletion of CTCF are shown in Fig. 1e.

Comment #13:

The effects of CTCF depletion on TADs/boundaries/loops have been examined in Zuin et al. (PMID: 24335803) and Rosa-Garrido et al. (PMID: 28802249), and these two studies seem to be good ones to reference within the context of the current manuscript.

Response #13:

We thank the reviewer for this comment, and the indicated references were added to the Introduction section of the revised manuscript.

Comment #14:

In Supplementary Figure 3A, the authors could probably show the same maximum values within the different groups of 4 heatmaps, to show more clearly that TAD structure is weakened with the CTCF knockout.

Response #14:

We thank the reviewer for this comment. In the revised manuscript, the Hi-C contact maps at each resolution are shown with the same maximum signal in the matrix.

Comment #15:

For a couple of columns in Supplemental Figure 4A-B, the numbers add up to greater than 100%. Please clarify these figures.

Response #15:

We apologize for the lack of clarity in these figures. We have now changed the labels in Supplementary Fig. 4a and 4b, as shown below.

Comment #16:

Some rationale for gene selection for Figure 3F would help the reader understand why each gene's interaction/transcription change is important for the resultant phenotype after LPS.

Response #16:

We thank the reviewer for raising this point. We believe that this comment is based on a misunderstanding that we may have caused due to inaccurate description of the legend of Fig. 3f. We have changed this legend in the revised manuscript as follows: “Log2-fold changes in RNA expression for typical genes representing each regulatory mode shown in (c),” not in (d).

The genes shown in Fig. 3f and 3g were selected as examples of six possible regulatory modes (Fig. 3c) controlling LPS-stimulated gene expression (Fig. 3e), categorized by different combinations of dynamic changes in loop strength and distal enhancer (DE) activity: (a) gain of loop to activation-inducible enhancer leads to increased mRNA expression of *Ptgs2*; (b) preformed loop with activation-inducible enhancer leads to increased mRNA expression of *Acod1*; (c) gain of loop to constitutive enhancer leads to increased mRNA expression of *Vash1*; (d) loss of loop from constitutive enhancer leads to decreased mRNA expression of *Tef*; (e) preformed loop with activation-repressed enhancer leads to decreased mRNA expression of *Foxred2*; and (f) loss of loop from activation-repressed enhancer leads to decreased mRNA expression of *Engase* (Fig. 3f and 3g).

Comment #17:

How do the authors define loops gained in CTCF knockout in Figure 4? Are they still examining HiChIP data?

Response #17:

We thank the reviewer for indicating the lack of clarity in our description of loops in Fig. 4. As in the case for LPS stimulation in Fig. 3, differential analysis of chromatin interactions between WT and KO BMDCs were performed using H3K27ac HiChIP data to define gained or lost H3K27ac loops. We have now added the MA plot for differential H3K27ac HiChIP loop analysis between WT and KO BMDCs as Supplementary Fig. 6c. We hope that this explanation clarifies the reviewer's point.

Supplementary Figure 6. (related to Figure 4) Gain of loops to already-active distal enhancers due to CTCF depletion leads to dysregulated gene expression c Volcano plot showing differential H3K27ac HiChIP loops between WT and KO BMDCs. The number of loops exhibiting >1.5-fold in WT (blue) or KO (red) BMDCs with a P-value <0.1 is indicated.

Comment #18:

Figure 4H should have an indication of the n for the experiment, and Figure 4I needs molecular weight markers and an indication of how many blots were performed.

Response #18:

We thank the reviewer for this suggestion. The number of biological replicates for Fig. 4h and 4i was described at the corresponding figure legend, and molecular weight markers were labeled in Fig. 4i.

Comment #19:

Supplemental Figure 5E does not show a $\log_2(\text{fold-change})$. Is this an RT-qPCR experiment?

Response #19:

We apologize that we did not describe the graph adequately. We now changed the legend

of Supplementary Fig. 6f (corresponding to Supplementary Fig. 5e in the original version of the manuscript).

Comment #20:

The analysis in Figure 5G is very useful to the reader and is a great example of how to dive deeper into Gene Ontology analyses. Are these all the genes in the “Regulation of NF- κ B activity” term? If not, it would be useful to see all of them together on the graph.

Response #20:

We thank the reviewer for this comment. All the genes in the “Regulation of NF- κ B activity” term were listed in Fig. 5g.

Comment #21:

Figure 6 could use extra labeling to indicate that Figure 6 I-J shows H3K27ac loops. A brief description of the LOLA algorithm would be useful here as well.

Response #21:

As suggested, we have now changed the legend for Fig. 6j and 6k from “H3K27ac loop” to “H3K27ac HiChIP loop”. We apologize for the lack of information regarding the LOLA algorithm, and we have now added a brief description about that in the Methods section of the revised manuscript.

Comment #22:

Why are the FDR significance thresholds different for the significant loops between the H3K27ac and RelA HiChIP experiments? Some commentary on this would provide deeper insight into the technical details of these experiments.

Response #22:

HiChIP is a method used to study chromatin contacts mediated by specific proteins, such as architectural proteins, cell type-specific transcription factors, or histone modifications. In this study, loop-calling for the RelA HiChIP experiment was performed using FitHiChIP with default parameter settings (FDR <0.01). However, we believe that more stringent criteria (FDR <10⁻⁵) were required for loop-calling for the H3K27ac HiChIP experiment given that H3K27ac histone modification, unlike transcription factors with sharp and narrow peaks, frequently generates broad peaks and that lots of noise interactions could be considered as significant loops connecting anchors with low-signal H3K27ac located in the broad peak area.

Comment #23:

Figures 7G-H are not called out in the main text.

Response #23:

We thank the reviewer for this comment. The figure labels were added in the main text as following: The 3D clique analysis of the *Il6* and *Il12a* loci also demonstrated that LPS stimulation drives the formation of the highly interacting spatial clusters between activation-inducible enhancers and promoters in WT BMDCs but not in CTCF-deficient BMDCs (Fig. 7g and 7h)

Again, we thank the reviewers for their kind consideration in reviewing and commenting on our manuscript and giving us the opportunity to respond to their valuable comments.

REFERENCES

- 1 Helft, J. *et al.* GM-CSF Mouse Bone Marrow Cultures Comprise a Heterogeneous Population of CD11c+MHCII+ Macrophages and Dendritic Cells. *Immunity* **42**, 1197-1211, doi:<https://doi.org/10.1016/j.immuni.2015.05.018> (2015).
- 2 Guilliams, M. & Malissen, B. A Death Notice for In-Vitro-Generated GM-CSF Dendritic Cells? *Immunity* **42**, 988-990, doi:<https://doi.org/10.1016/j.immuni.2015.05.020> (2015).
- 3 Menges, M. *et al.* IL-4 supports the generation of a dendritic cell subset from murine bone marrow with altered endocytosis capacity. *J Leukoc Biol* **77**, 535-543, doi:10.1189/jlb.0804473 (2005).
- 4 Lutz, Manfred B., Inaba, K., Schuler, G. & Romani, N. Still Alive and Kicking: In-Vitro-Generated GM-CSF Dendritic Cells! *Immunity* **44**, 1-2, doi:<https://doi.org/10.1016/j.immuni.2015.12.013> (2016).
- 5 Lu, L., McCaslin, D., Starzl, T. E. & Thomson, A. W. Bone marrow-derived dendritic cell progenitors (NLDC 145+, MHC class II+, B7-1dim, B7-2-) induce alloantigen-specific hyporesponsiveness in murine T lymphocytes. *Transplantation* **60**, 1539-1545, doi:10.1097/00007890-199560120-00028 (1995).
- 6 Son, Y.-I. *et al.* A novel bulk-culture method for generating mature dendritic cells from mouse bone marrow cells. *Journal of Immunological Methods* **262**, 145-157, doi:[https://doi.org/10.1016/S0022-1759\(02\)00013-3](https://doi.org/10.1016/S0022-1759(02)00013-3) (2002).
- 7 Lutz, M. B. IL-3 in dendritic cell development and function: a comparison with GM-CSF and IL-4. *Immunobiology* **209**, 79-87, doi:<https://doi.org/10.1016/j.imbio.2004.03.001> (2004).
- 8 Xu, Y., Zhan, Y., Lew, A. M., Naik, S. H. & Kershaw, M. H. Differential Development of Murine Dendritic Cells by GM-CSF versus Flt3 Ligand Has Implications for Inflammation and Trafficking. *The Journal of Immunology* **179**, 7577, doi:10.4049/jimmunol.179.11.7577 (2007).
- 9 Na, Y. R., Jung, D., Gu, G. J. & Seok, S. H. GM-CSF Grown Bone Marrow Derived Cells Are Composed of Phenotypically Different Dendritic Cells and Macrophages. *Mol Cells* **39**, 734-741, doi:10.14348/molcells.2016.0160 (2016).
- 10 Helft, J. *et al.* Alive but Confused: Heterogeneity of CD11c(+) MHC Class II(+) Cells in GM-CSF Mouse Bone Marrow Cultures. *Immunity* **44**, 3-4, doi:10.1016/j.immuni.2015.12.014 (2016).
- 11 Hyle, J. *et al.* Acute depletion of CTCF directly affects MYC regulation through loss of enhancer-promoter looping. *Nucleic Acids Res* **47**, 6699-6713, doi:10.1093/nar/gkz462 (2019).
- 12 Lee, R. *et al.* CTCF-mediated chromatin looping provides a topological framework for the formation of phase-separated transcriptional condensates. *Nucleic Acids Res* **50**, 207-226, doi:10.1093/nar/gkab1242 (2022).
- 13 Nora, E. P. *et al.* Targeted Degradation of CTCF Decouples Local Insulation of Chromosome Domains from Genomic Compartmentalization. *Cell* **169**, 930-944.e922,

- doi:10.1016/j.cell.2017.05.004 (2017).
- 14 Shin, H.-M. *et al.* Inhibitory action of novel aromatic diamine compound on lipopolysaccharide-induced nuclear translocation of NF- κ B without affecting I κ B degradation. *FEBS Letters* **571**, 50-54, doi:<https://doi.org/10.1016/j.febslet.2004.06.056> (2004).
- 15 Swift, S., Lorens, J., Achacoso, P. & Nolan, G. P. Rapid production of retroviruses for efficient gene delivery to mammalian cells using 293T cell-based systems. *Curr Protoc Immunol* **Chapter 10**, Unit 10.17C, doi:10.1002/0471142735.im1017cs31 (2001).
- 16 Wu, Y. *et al.* FOXP3 controls regulatory T cell function through cooperation with NFAT. *Cell* **126**, 375-387, doi:10.1016/j.cell.2006.05.042 (2006).
- 17 Nora, E. P. *et al.* Molecular basis of CTCF binding polarity in genome folding. *Nat Commun* **11**, 5612, doi:10.1038/s41467-020-19283-x (2020).
- 18 Nakahashi, H. *et al.* A genome-wide map of CTCF multivalency redefines the CTCF code. *Cell Rep* **3**, 1678-1689, doi:10.1016/j.celrep.2013.04.024 (2013).
- 19 Zhao, D. *et al.* Molecular identification of a major retinoic-acid-synthesizing enzyme, a retinaldehyde-specific dehydrogenase. *Eur J Biochem* **240**, 15-22, doi:10.1111/j.1432-1033.1996.0015h.x (1996).
- 20 Coombes, J. L. *et al.* A functionally specialized population of mucosal CD103⁺ DCs induces Foxp3⁺ regulatory T cells via a TGF- β and retinoic acid-dependent mechanism. *J Exp Med* **204**, 1757-1764, doi:10.1084/jem.20070590 (2007).
- 21 Kwok, S.-K. *et al.* Retinoic Acid Attenuates Rheumatoid Inflammation in Mice. *The Journal of Immunology* **189**, 1062, doi:10.4049/jimmunol.1102706 (2012).
- 22 Sun, C. M. *et al.* Small intestine lamina propria dendritic cells promote de novo generation of Foxp3 T reg cells via retinoic acid. *J Exp Med* **204**, 1775-1785, doi:10.1084/jem.20070602 (2007).
- 23 Mallette, F. A., Gaumont-Leclerc, M. F., Huot, G. & Ferbeyre, G. Myc down-regulation as a mechanism to activate the Rb pathway in STAT5A-induced senescence. *J Biol Chem* **282**, 34938-34944, doi:10.1074/jbc.M707074200 (2007).
- 24 Onishi, M. *et al.* Identification and characterization of a constitutively active STAT5 mutant that promotes cell proliferation. *Mol Cell Biol* **18**, 3871-3879, doi:10.1128/mcb.18.7.3871 (1998).
- 25 Luckheeram, R. V., Zhou, R., Verma, A. D. & Xia, B. CD4⁺ T cells: differentiation and functions. *Clin Dev Immunol* **2012**, 925135, doi:10.1155/2012/925135 (2012).
- 26 Zhu, J., Yamane, H. & Paul, W. E. Differentiation of effector CD4 T cell populations (*). *Annu Rev Immunol* **28**, 445-489, doi:10.1146/annurev-immunol-030409-101212 (2010).
- 27 Hsieh, C. S. *et al.* Development of TH1 CD4⁺ T cells through IL-12 produced by Listeria-induced macrophages. *Science* **260**, 547-549, doi:10.1126/science.8097338 (1993).
- 28 Bettelli, E. *et al.* Reciprocal developmental pathways for the generation of pathogenic effector TH17 and regulatory T cells. *Nature* **441**, 235-238, doi:10.1038/nature04753 (2006).

- 29 Chang, S., Collins, P. L. & Aune, T. M. T-bet dependent removal of Sin3A-histone deacetylase complexes at the *Ifng* locus drives Th1 differentiation. *J Immunol* **181**, 8372-8381, doi:10.4049/jimmunol.181.12.8372 (2008).
- 30 Durant, L. *et al.* Diverse targets of the transcription factor STAT3 contribute to T cell pathogenicity and homeostasis. *Immunity* **32**, 605-615, doi:10.1016/j.immuni.2010.05.003 (2010).
- 31 Renaude, E. *et al.* The Fate of Th17 Cells is Shaped by Epigenetic Modifications and Remodeled by the Tumor Microenvironment. *Int J Mol Sci* **21**, doi:10.3390/ijms21051673 (2020).

REVIEWER COMMENTS

Reviewer #1 (Remarks to the Author):

The authors have altered the terminology to an acceptable form, but the manuscript needs to be curated for a few places where the old terminology of "DCs" can still be found.

Beyond that, the authors have addressed the remainder of my concerns

Reviewer #2 (Remarks to the Author):

The authors have adequately addressed this reviewer's comments. The manuscript is now much stronger, as evinced by the new phenotyping figure and clarification (Supplemental Figure 1), the STAT5 over expression and transcription measurements (Supplemental Figure 7), and the new HiChIP experiment with the JSH-23 treatment (Figure 8). Additional clarifications have also made the manuscript more accessible to a broader audience. Well done.

Point-by-point responses to the reviewers' comments

COMMENTS FROM REVIEWER #1

The authors have altered the terminology to an acceptable form, but the manuscript needs to be curated for a few places where the old terminology of "DCs" can still be found. Beyond that, the authors have addressed the remainder of my concerns

Response to reviewer #1:

As requested, we have changed the terminology of “DCs” to “BMDCs” and “dendritic cells” to “bone marrow-derived dendritic cells” in the title, abstract, introduction, and discussion sections. Specifically, we have changed the title of our manuscript from “CTCF is essential to establish three-dimensional enhancer network underlying the inflammatory response of dendritic cells” to “CTCF controls three-dimensional enhancer network underlying the inflammatory response of bone marrow-derived dendritic cells”. We hope that this curation clarifies the reviewer’s point.

COMMENTS FROM REVIEWER #2

The authors have adequately addressed this reviewer's comments. The manuscript is now much stronger, as evinced by the new phenotyping figure and clarification (Supplemental Figure 1), the STAT5 over expression and transcription measurements (Supplemental Figure 7), and the new HiChIP experiment with the JSH-23 treatment (Figure 8). Additional clarifications have also made the manuscript more accessible to a broader audience. Well done.

Response to reviewer #2:

We sincerely thank the reviewer for the appreciation of our work and the positive feedback.

Again, we thank the reviewers for their kind consideration in reviewing and commenting on our manuscript and giving us the opportunity to respond to their valuable comments.